# LCOT: LINEAR CIRCULAR OPTIMAL TRANSPORT

**Rocío Díaz Martín**[*1], **Ivan Medri**[*2], **Yikun Bai**[*3], **Xiran Liu**[3], **Kangbai Yan**[1],
**Gustavo K. Rohde** [†4], **Soheil Kolouri** [‡3]

[1]Department of Mathematics, Vanderbilt University, Nashville, TN 37240, USA.
[2]Department of Computer Science, Tennessee State University, Nashville, TN 37209, USA.
[3]Department of Computer Science, Vanderbilt University, Nashville, TN 37240, USA.
[4]Department of Biomedical Engineering, Department of Electrical and Computer Engineering,
 University of Virginia, Charlottesville, VA 22908, USA.
[1]{rocio.p.diaz.martin,kangbai.yan}@vanderbilt.edu
[3]{yikun.bai,xinran.liu,soheil.kolouri}@vanderbilt.edu
[2]imedri@tnstate.edu, [4]gustavo@virginia.edu

## ABSTRACT

The optimal transport problem for measures supported on non-Euclidean spaces has recently gained ample interest in diverse applications involving representation learning. In this paper, we focus on circular probability measures, i.e., probability measures supported on the unit circle, and introduce a new computationally efficient metric for these measures, denoted as Linear Circular Optimal Transport (LCOT). The proposed metric comes with an explicit linear embedding that allows one to apply Machine Learning (ML) algorithms to the embedded measures and seamlessly modify the underlying metric for the ML algorithm to LCOT. We show that the proposed metric is rooted in the Circular Optimal Transport (COT) and can be considered the linearization of the COT metric with respect to a fixed reference measure. We provide a theoretical analysis of the proposed metric and derive the computational complexities for pairwise comparison of circular probability measures. Lastly, through a set of numerical experiments, we demonstrate the benefits of LCOT in learning representations of circular measures.

## 1 INTRODUCTION

Optimal transport (OT) (Villani, 2009; Peyré et al., 2019) is a mathematical framework that seeks the most efficient way of transforming one probability measure into another. The OT framework leads to a geometrically intuitive and robust metric on the set of probability measures, referred to as the Wasserstein distance. It has become an increasingly popular tool in machine learning, data analysis, and computer vision (Kolouri et al., 2017; Khamis et al., 2023). OT's applications encompass generative modeling (Arjovsky et al., 2017; Tolstikhin et al., 2017; Kolouri et al., 2018), domain adaptation (Courty et al., 2017; Damodaran et al., 2018), transfer learning (Alvarez-Melis & Fusi, 2020; Liu et al., 2022), supervised learning (Frogner et al., 2015), clustering (Ho et al., 2017), image and pointcloud registration (Haker et al., 2004; Bai et al., 2022; Le et al., 2023), and even inverse problems (Mukherjee et al., 2021), among others. Recently, there has been an increasing interest in OT for measures supported on manifolds (Bonet et al., 2023; Sarrazin & Schmitzer, 2023). This surging interest is primarily due to: 1) real-world data is often supported on a low-dimensional manifold embedded in larger-dimensional Euclidean spaces, and 2) many applications inherently involve non-Euclidean geometry, e.g., geophysical data or cortical signals in the brain.

In this paper, we are interested in efficiently comparing probability measures supported on the unit circle, aka circular probability measures, using the optimal transport framework. Such probability measures, with their densities often represented as circular/rose histograms, are prevalent in many applications, from computer vision and signal processing domains to geology and astronomy. For instance, in classic computer vision, the color content of an image can be accounted for by its hue

---

[*]These authors contributed equally to this work.
[†]Acknowledges support from ONR N000142212505, and NIH GM130825.
[‡]Acknowledges partial support from the Defense Advanced Research Projects Agency (DARPA) under Contract No. HR00112190135 and HR00112090023, and the Wellcome LEAP Foundation.

in the HSV space, leading to one-dimensional circular histograms. Additionally, local image/shape descriptors are often represented via circular histograms, as evidenced in classic computer vision papers like SIFT (Lowe, 2004) and ShapeContext (Belongie et al., 2000). In structural geology, the orientation of rock formations, such as bedding planes, fault lines, and joint sets, can be represented via circular histograms Twiss & Moores (1992). In signal processing, circular histograms are commonly used to represent the phase distribution of periodic signals (Levine et al., 2002). Additionally, a periodic signal can be normalized and represented as a circular probability density function (PDF).

Notably, a large body of literature exists on circular statistics (Jammalamadaka & SenGupta, 2001). More specific to our work, however, are the seminal works of Delon et al. (2010) and Rabin et al. (2011), which provide a thorough study of the OT problem and transportation distances on the circle (see also Cabrelli & Molter (1998)). OT on circles has also been recently revisited in various papers (Hundrieser et al., 2022; Bonet et al., 2023; Beraha & Pegoraro, 2023; Quellmalz et al., 2023), further highlighting the topic's timeliness. Unlike OT on the real line, generally, the OT problem between probability measures defined on the circle does not have a closed-form solution. This stems from the intrinsic metric on the circle and the fact that there are two paths between any pair of points on a circle (i.e., clockwise and counter-clockwise). Interestingly, however, when one of the probability measures is the Lebesgue measure, i.e., the uniform distribution, the 2-Wasserstein distance on the circle has a closed-form solution, which we will discuss in the Background section.

We present the Linear Circular OT (LCOT), a new transport-based distance for circular probability measures. By leveraging the closed-form solution of the circular 2-Wasserstein distance between each distribution and the uniform distribution on the circle, our method sidesteps the need for optimization. Concisely, we determine the Monge maps that push the uniform distribution to each input measure using the closed-form solution, then set the distance between the input measures based on the disparities between their respective Monge maps. Our approach draws parallels with the Linear Optimal Transport (LOT) framework proposed by Wang et al. (2013) and can be seen as an extension of the cumulative distribution transform (CDT) presented by (Park et al., 2018) to circular probability measures (see also, Aldroubi et al. (2022; 2021)). The idea of linearized (unbalanced) optimal transport was also studied recently in various works (Cai et al., 2022; Moosmüller & Cloninger, 2023; Sarrazin & Schmitzer, 2023; Cloninger et al., 2023). From a geometric perspective, we provide explicit logarithmic and exponential maps between the space of probability measures on the unit circle and the tangent space at a reference measure (e.g., the Lebesgue measure) Wang et al. (2013); Cai et al. (2022); Sarrazin & Schmitzer (2023). Then, we define our distance in this tangent space, giving rise to the terminology 'Linear' Circular OT. The logarithmic map provides a linear embedding for the LCOT distance, while the exponential map inverts this embedding. We provide a theoretical analysis of the proposed metric, LCOT, and demonstrate its utility in various problems.

**Contributions.** Our specific contributions in this paper include 1) proposing a computationally efficient metric for circular probability measures, 2) providing a theoretical analysis of the proposed metric, including its computational complexity for pairwise comparison of a set of circular measures, and 3) demonstrating the robustness of the proposed metric in manifold learning, measure interpolation, and clustering/classification of probability measures.

## 2 BACKGROUND

### 2.1 CIRCLE SPACE

The unit circle $\mathbb{S}^1$ can be defined as the quotient space

$$\mathbb{S}^1 = \mathbb{R}/\mathbb{Z} = \{\{x + n : n \in \mathbb{Z}\} : x \in [0, 1)\}.$$

The above definition is equivalent to $[0, 1]$ under the identification $0 = 1$. For the sake of simplicity in this article, we treat them as indistinguishable. Furthermore, we adopt a parametrization of the circle as $[0, 1)$, designating the North Pole as $0$ and adopting a clockwise orientation. This will serve as our *canonical* parametrization.

Let $|\cdot|$ denote the absolute value on $\mathbb{R}$. With the aim of avoiding any confusion, when necessary, we will denote it by $|\cdot|_{\mathbb{R}}$. Then, a metric on $\mathbb{S}^1$ can be defined as

$$|x - y|_{\mathbb{S}^1} := \min\{|x - y|_{\mathbb{R}}, 1 - |x - y|_{\mathbb{R}}\}, \qquad x, y \in [0, 1)$$

or, equivalently, as

$$|x - y|_{\mathbb{S}^1} := \min_{k \in \mathbb{Z}} |x - y + k|_{\mathbb{R}},$$

where for the second formula $x, y \in \mathbb{R}$ are understood as representatives of two classes of equivalence in $\mathbb{R}/\mathbb{Z}$, but these two representatives $x, y$ do not need to belong to $[0, 1)$. It turns out that such a metric defines a geodesic distance: It is the smaller of the two arc lengths between the points $x$, $y$ along the circumference (cf. Jammalamadaka & SenGupta (2001), where here we parametrize angles between 0 and 1 instead of between 0 and $2\pi$. Besides, the circle $\mathbb{S}^1$ can be endowed with a group structure. Indeed, as the quotient space $\mathbb{R}/\mathbb{Z}$ it inherits the addition from $\mathbb{R}$ modulo $\mathbb{Z}$. Equivalently, for any $x, y \in [0, 1)$, we can define the operations $+, -$ as

$$(x, y) \mapsto \begin{cases} x \pm y, & \text{if } x \pm y \in [0, 1) \\ x \pm y \mp 1, & \text{otherwise.} \end{cases} \tag{1}$$

## 2.2 DISTRIBUTIONS ON THE CIRCLE

Regarded as a set, $\mathbb{S}^1$ can be identified with $[0, 1)$. Thus, signals over $\mathbb{S}^1$ can be interpreted as 1-periodic functions on $\mathbb{R}$. More generally, every measure $\mu \in \mathcal{P}(\mathbb{S}^1)$ can be regarded as a measure on $\mathbb{R}$ by

$$\mu(A + n) := \mu(A), \qquad \text{for every } A \subseteq [0, 1) \text{ Borel subset, and } n \in \mathbb{Z}. \tag{2}$$

Then, its *cumulative distribution function*, denoted by $F_\mu$, is defined as

$$F_\mu(y) := \mu([0, y)) = \int_0^y d\mu, \qquad \forall y \in [0, 1) \tag{3}$$

and can be extended to a function on $\mathbb{R}$ by

$$F_\mu(y + n) := F_\mu(y) + n, \qquad \forall y \in [0, 1), \, n \in \mathbb{Z}. \tag{4}$$

Figure 2 shows the concept of $F_\mu$ and its extension to $\mathbb{R}$.

In the rest of this article, we do not distinguish between the definition of measures on $\mathbb{S}^1$ or their periodic extensions into $\mathbb{R}$, as well as between their CDFs or their extended CDFs into $\mathbb{R}$.

**Definition 2.1.** *[Cumulative distribution function with respect to a reference point] Let $\mu \in \mathcal{P}(\mathbb{S}^1)$, and consider a reference point $x_0 \in \mathbb{S}^1$. Assume that $\mathbb{S}^1$ is identified as $[0, 1)$ according to our canonical parametrization. By abuse of notation, also denote by $x_0$ the point in $[0, 1)$ that corresponds to the given reference point when considering the canonical parametrization. We define*

$$F_{\mu, x_0}(y) := F_\mu(x_0 + y) - F_\mu(x_0).$$

The reference point $x_0$ can be considered as the "origin" for parametrizing the circle as $[0, 1)$ starting from $x_0$. That is, $x_0$ will correspond to 0, and from there, we move according to the clockwise orientation. Thus, we can think of $x_0$ in the above definition as a "cutting point": A point where we cut $\mathbb{S}^1$ into a line by $x_0$ and so we can unroll PDFs and CDFs over the circle into $\mathbb{R}$. See Figures 1 and 2.

Besides, note that $F_{\mu, x_0}(0) = 0$ and $F_{\mu, x_0}(1) = 1$ by the 1-periodicity of $\mu$. This is to emphasize that in the new system of coordinates, or in the new parametrization of $\mathbb{S}^1$ as $[0, 1)$ starting from $x_0$, the new origin $x_0$ plays

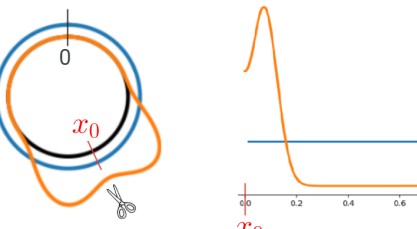

Figure 1: Visualization of densities (blue and orange) on $\mathbb{S}^1$ and after unrolling them to $[0, 1)$ by considering a cutting point $x_0$. The blue density is the uniform distribution on $\mathbb{S}^1$, represented as having height 1 over the unit circle in black.

the role of 0. Finally, notice that if $x_0$ is the North Pole, which corresponds to 0 in the canonical parametrization of the circle, then $F_{\mu, x_0} = F_\mu$.

**Definition 2.2.** *The quantile function $F_{\mu, x_0}^{-1} : [0, 1] \to [0, 1]$ is defined as $F_{\mu, x_0}^{-1}(y) := \inf\{x : F_{\mu, x_0}(x) > y\}$.*

## 2.3 OPTIMAL TRANSPORT ON THE CIRCLE

### 2.3.1 PROBLEM SETUP

Given $\mu, \nu \in \mathcal{P}(\mathbb{S}^1)$, let $c(x, y) := h(|x - y|_{\mathbb{S}^1})$ be the cost of transporting a unit mass from $x$ to $y$ on the circle, where $h : \mathbb{R} \to \mathbb{R}_+$ is a convex increasing function. The Circular Optimal Transport

cost between $\mu$ and $\nu$ is defined as

$$COT_h(\mu, \nu) := \inf_{\gamma \in \Gamma(\mu,\nu)} \int_{\mathbb{S}^1 \times \mathbb{S}^1} c(x,y)\, d\gamma(x,y), \tag{5}$$

where $\Gamma(\mu, \nu)$ is the set of all transport plans from $\mu$ to $\nu$, that is, $\gamma \in \Gamma(\mu, \nu)$ is such that $\gamma \in \mathcal{P}(\mathbb{S}^1 \times \mathbb{S}^1)$ having first and second marginals $\mu$ and $\nu$, respectively. There always exists a minimizer $\gamma^*$ of 5, and it is called a Kantorovich optimal plan (see, for example, Santambrogio (2015, Th. 1.4)).

When $h(x) = |x|^p$, for $1 \leq p < \infty$, we denote $COT_h(\cdot, \cdot) = COT_p(\cdot, \cdot)$, and $COT_p(\cdot, \cdot)^{1/p}$ defines a distance on $\mathcal{P}(\mathbb{S}^1)$. In general,

$$COT_h(\mu, \nu) \leq \inf_{M:\, M_\#\mu=\nu} \int_{\mathbb{S}^1} h(|M(x) - x|_{\mathbb{S}^1})\, d\mu(x), \tag{6}$$

and a minimizer $M^* : \mathbb{S}^1 \to \mathbb{S}^1$ of the right-hand side of 6, among all maps $M$ that pushforward $\mu$ to $\nu$ [1], might not exist. In this work, we will consider the cases where a minimizer $M^*$ does exist, for example, when the reference measure $\mu$ is absolutely continuous with respect to the Lebesgue measure on $\mathbb{S}^1$ (see McCann (2001); Santambrogio (2015)). In these scenarios, such map $M^*$ is called an optimal transportation map or a Monge map. Moreover, as $\mu, \nu \in \mathcal{P}(\mathbb{S}^1)$ can be regarded as measures on $\mathbb{R}$ according to equation 2, we can work with transportation maps $M : \mathbb{R} \to \mathbb{R}$ that are 1-periodic functions satisfying $M_\#\mu = \nu$.

**Proposition 2.3.** *Two equivalent formulations of $COT_h$ are the following:*

$$COT_h(\mu, \nu) = \inf_{x_0 \in [0,1)} \int_0^1 h(|F_{\mu,x_0}^{-1}(x) - F_{\nu,x_0}^{-1}(x)|_\mathbb{R})\, dx \tag{7}$$

$$= \inf_{\alpha \in \mathbb{R}} \int_0^1 h(|F_\mu^{-1}(x) - F_\nu^{-1}(x-\alpha)|_\mathbb{R})\, dx. \tag{8}$$

*When there exist minimizers $x_{cut}$ and $\alpha_{\mu,\nu}$ of equation 7 and equation 8, respectively, the relation between them is given by*

$$\alpha_{\mu,\nu} = F_\mu(x_{cut}) - F_\nu(x_{cut}). \tag{9}$$

*Moreover, if $\mu = Unif(\mathbb{S}^1)$ and $h(x) = |x|^2$, it can be verified that $\alpha_{\mu,\nu}$ is the antipodal of $\mathbb{E}(\nu)$, i.e.,*

$$\alpha_{\mu,\nu} = x_{cut} - F_\nu(x_{cut}) = \mathbb{E}(\nu) - 1/2. \tag{10}$$

The proof of equation 8 in Proposition 2.3 is provided in Delon et al. (2010) for the optimal coupling for any pair of probability measures on $\mathbb{S}^1$. For the particular and enlightening case of discrete probability measures on $\mathbb{S}^1$, we refer the reader to Rabin et al. (2011). In that article, equation 7 is introduced. Finally, equation 10 is given for example in Bonet et al. (2023, Proposition 1).

---

[1] The pushforward $M_\#\mu$ is defined by the change of variables $\int \varphi(y)\, dM_\#\mu(y) := \int \varphi(M(x))\, d\mu(x)$, for every continuous function $\varphi : \mathbb{S}^1 \to \mathbb{C}$.

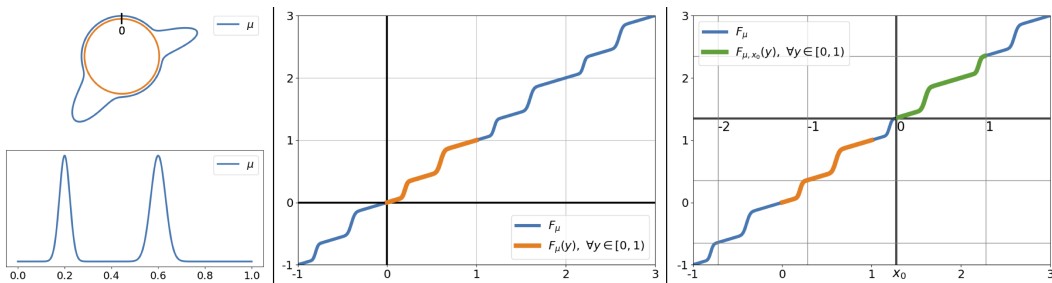

Figure 2: Left: The density of a probability measure, $\mu$. Middle: visualization of the periodic extension to $\mathbb{R}$ of a CDF, $F_\mu$, of measure $\mu$ on $[0, 1) \sim \mathbb{S}^1$. Right: Visualization of $F_{\mu,x_0}$ given in Definition 2.1, where the parameterization of the circle is changed; now, the origin 0 is the point $x_0$.

Proposition 2.3 allows us to see the optimization problem of transporting measures supported on the circle as an optimization problem on the real line by looking for the best "cutting point" so that the circle can be unrolled into the real line by 1-periodicity.

**Remark 2.4.** *In general, if $h$ is strictly convex, the minimizer of equation 8 is unique (see Appendix A.2), but there can be multiple minimizers for equation 7 (see Figure 8 in Appendix A.2 and equation 9). However, when a minimizer $x_{cut}$ of equation 7 exists, it will lead to the optimal transportation displacement on a circular domain (see Section 2.3.2 below).*

### 2.3.2 A CLOSED-FORM FORMULA FOR THE OPTIMAL CIRCULAR DISPLACEMENT

Let $x_{cut}$ be a minimizer of equation 7, that is,

$$COT_h(\mu, \nu) = \int_0^1 h(|F_{\mu,x_{cut}}^{-1}(x) - F_{\nu,x_{cut}}^{-1}(x)|_{\mathbb{R}}) \, dx. \tag{11}$$

From equation 11, one can now emulate the Optimal Transport Theory on the real line (see, for e.g., Santambrogio (2015)): The point $x_{cut}$ provides a reference where one can "cut" the circle. Subsequently, computing the optimal transport between $\mu$ and $\nu$ boils down to solving an optimal transport problem between two distributions on the real line.

We consider the parametrization of $\mathbb{S}^1$ as $[0, 1)$ by setting $x_{cut}$ as the origin and moving along the clockwise orientation. Let us use the notation $\widetilde{x} \in [0, 1)$ for the points given by such parametrization, and the notation $x \in [0, 1)$ for the canonical parametrization. That is, the change of coordinates from the two parametrizations is given by $x = \widetilde{x} + x_{cut}$. Then, if $\mu$ does not give mass to atoms, by equation 11 and the classical theory of Optimal Transport on the real line, the optimal transport map (Monge map) that takes a point $\widetilde{x}$ to a point $\widetilde{y}$ is given by

$$F_{\nu,x_{cut}}^{-1} \circ F_{\mu,x_{cut}}(\widetilde{x}) = \widetilde{y} \tag{12}$$

That is, 12 defines a circular optimal transportation map from $\mu$ to $\nu$ written in the parametrization that establishes $x_{cut}$ as the "origin." If we want to refer everything to the original labeling of the circle, that is, if we want to write equation 12 with respect to the canonical parametrization, we need to change coordinates

$$\begin{cases} \widetilde{x} = & x - x_{cut} \\ \widetilde{y} = & y - x_{cut} \end{cases}. \tag{13}$$

Therefore, a closed-form formula for an optimal circular transportation map in the canonical coordinates is given by

$$M_\mu^\nu(x) := F_{\nu,x_{cut}}^{-1} \circ F_{\mu,x_{cut}}(x - x_{cut}) + x_{cut} = y, \qquad x \in [0, 1), \tag{14}$$

and the corresponding optimal circular transport displacement that takes $x$ to $y$ is

$$M_\mu^\nu(x) - x = F_{\nu,x_{cut}}^{-1} \circ F_{\mu,x_{cut}}(x - x_{cut}) - (x - x_{cut}), \qquad x \in [0, 1). \tag{15}$$

In summary, we condense the preceding discussion in the following result. The proof is provided in Appendix A.1. While the result builds upon prior work, drawing from Bonet et al. (2023); Rabin et al. (2011); Santambrogio (2015), it offers an explicit formula for the optimal Monge map.

**Theorem 2.5.** *Let $\mu, \nu \in \mathcal{P}(\mathbb{S}^1)$. Assume that $\mu$ is absolutely continuous with respect to the Lebesgue measure on $\mathbb{S}^1$ (that is, it does not give mass to atoms).*

1. *If $x_{cut}$ is a minimizer of equation 7, then equation 14 defines an optimal circular transportation map. (We will use the notation $M_\mu^\nu$ for the Monge map from $\mu$ to $\nu$.)*

2. *If $\alpha_{\mu,\nu}$ minimizes equation 8, then*

$$M_\mu^\nu(x) = F_\nu^{-1}(F_\mu(x) - \alpha_{\mu,\nu}) \tag{16}$$

3. *If $x_0, x_1$ are two minimizers of equation 7, then*
$$F_{\nu,x_0}^{-1} \circ F_{\mu,x_0}(x - x_0) + x_0 = F_{\nu,x_1}^{-1} \circ F_{\mu,x_1}(x - x_1) + x_1 \qquad \forall\, x \in [0, 1).$$

4. *If the cost function $h$ is strictly convex, the optimal map defined by the formula equation 14 is unique. (The uniqueness is as functions on $\mathbb{S}^1$, or as functions on $\mathbb{R}$ up to modulo $\mathbb{Z}$).*

5. *If also $\nu$ does not give mass to atoms, then $(M_\mu^\nu)^{-1} = M_\nu^\mu$.*

Having established the necessary background, we are now poised to introduce our proposed metric. In the subsequent section, we present the Linear Circular Optimal Transport (LCOT) metric.

## 3 METHOD

### 3.1 LINEAR CIRCULAR OPTIMAL TRANSPORT EMBEDDING (LCOT)

By following the footsteps ofWang et al. (2013), starting from the COT framework, we will define an embedding for circular measures by computing the optimal displacement from a fixed reference measure. Then, the $L^p$-distance on the embedding space defines a new distance between circular measures (Theorem 3.6 below).

**Definition 3.1** (LCOT Embedding). *For a fixed reference measure $\mu \in \mathcal{P}(\mathbb{S}^1)$ that is absolutely continuous with respect to the Lebesgue measure on $\mathbb{S}^1$, we define the Linear Circular Optimal Transport (LCOT) Embedding of a target measure $\nu \in \mathcal{P}(\mathbb{S}^1)$ with respect to the cost $COT_h(\cdot, \cdot)$, for a strictly convex increasing function $h : \mathbb{R} \to \mathbb{R}_+$, by*

$$\widehat{\nu}^{\mu,h}(x) := F_{\nu,x_{cut}}^{-1}(F_\mu(x - x_{cut})) - (x - x_{cut}) = F_\nu^{-1}(F_\mu(x) - \alpha_{\mu,\nu}) - x, \quad x \in [0,1), \quad (17)$$

*where $x_{cut}$ is any minimizer of equation 7 and $\alpha_{\mu,\nu}$ is the minimizer of equation 8.*

The LCOT-Embedding corresponds to the optimal (circular) displacement that comes from the problem of transporting the reference measure $\mu$ to the given target measure $\nu$ with respect to a general cost $COT_h(\cdot, \cdot)$ (see equation 16 from Theorem 2.5 and equation 15).

**Definition 3.2** (LCOT discrepancy). *Under the settings of Definition 3.1, we define the LCOT-discrepancy by*

$$LCOT_{\mu,h}(\nu_1, \nu_2) := \int_0^1 h\left(\min_{k\in\mathbb{Z}}\{|\widehat{\nu_1}^{\mu,h}(t) - \widehat{\nu_2}^{\mu,h}(t) + k|_{\mathbb{R}}\}\right) d\mu(t), \quad \forall \nu_1, \nu_2 \in \mathcal{P}(\mathbb{S}^1).$$

*In particular, when $h(\cdot) = |\cdot|^p$, for $1 < p < \infty$, we define the $LCOT_{\mu,p}$ distance as*

$$LCOT_{\mu,p}(\nu_1, \nu_2) := \|\widehat{\nu_1}^{\mu,h} - \widehat{\nu_2}^{\mu,h}\|_{L^p(\mathbb{S}^1, d\mu)}^p = \int_0^1 \left(\min_{k\in\mathbb{Z}}\{|\widehat{\nu_1}^{\mu,h}(t) - \widehat{\nu_2}^{\mu,h}(t) + k|_{\mathbb{R}}\}\right)^p d\mu(t)$$

$$where \quad L^p(\mathbb{S}^1, d\mu) := \{f : \mathbb{S}^1 \to \mathbb{R} \mid \quad \|f\|_{L^p(\mathbb{S}^1, d\mu)} := \left(\int_{\mathbb{S}^1} |f(t)|_{\mathbb{S}^1}^p \, d\mu(t)\right)^{1/p} < \infty\}. \quad (18)$$

*If $\mu = Unif(\mathbb{S}^1)$, we use the notation $L^p(\mathbb{S}^1) := L^p(\mathbb{S}^1, d\mu)$.*

The embedding $\nu \mapsto \widehat{\nu}$ as outlined by equation 17 is consistent with the definition of the Logarithm function given in (Sarrazin & Schmitzer, 2023, Definition 2.7) (we also refer to Wang et al. (2013) for the LOT framework). However, the emphasis of the embedding in this paper is on computational efficiency, and a closed-form solution is provided. Additional details are available in Appendix A.6.

**Remark 3.3.** *If the reference measure is $\mu = Unif(\mathbb{S}^1)$, given a target measure $\nu \in \mathcal{P}(\mathbb{S}^1)$, we denote the LCOT-Embedding $\widehat{\nu}^{\mu,h}$ of $\nu$ with respect to the cost $COT_2(\cdot, \cdot)$ (i.e., $h(x) = |x|^2$) simply by $\widehat{\nu}$. Due to Theorem 2.5 and equation 10, the expression 17 reduces to*

$$\widehat{\nu}(x) := F_\nu^{-1}\left(x - \mathbb{E}(\nu) + \frac{1}{2}\right) - x, \qquad x \in [0,1). \quad (19)$$

*In this setting, we denote $LCOT_{\mu,h}(\cdot, \cdot)$ simply by $LCOT(\cdot, \cdot)$. That is, given $\nu_1, \nu_2 \in \mathcal{P}(\mathbb{S}^1)$,*

$$LCOT(\nu_1, \nu_2) := \|\widehat{\nu_1} - \widehat{\nu_2}\|_{L^2(\mathbb{S}^1)}^2 = \int_0^1 \left(\min_{k\in\mathbb{Z}}\{|\widehat{\nu_1}(t) - \widehat{\nu_2}(t) + k|_{\mathbb{R}}\}\right)^2 dt. \quad (20)$$

*All our experiments are performed using the embedding $\widehat{\nu}$ given by 19 due to the robustness of the closed-form formula 10 for the minimizer $\alpha_{\mu,\nu}$ of equation 8 when $h(x) = |x|^2$ and $\mu = Unif(\mathbb{S}^1)$.*

**Remark 3.4.** *Let $\mu \in \mathcal{P}(\mathbb{S}^1)$ be absolutely continuous with respect to the Lebesgue measure on $\mathbb{S}^1$, and $h : \mathbb{R} \to \mathbb{R}_+$ a strictly convex increasing function. Given $\nu \in \mathcal{P}(\mathbb{S}^1)$.*

$$COT_h(\mu, \nu) = \int_0^1 h\left(|\widehat{\nu}^{\mu,h}(t)|_{\mathbb{S}^1}\right) dt = \int_0^1 h\left(|\widehat{\nu}^{\mu,h}(t) - \widehat{\mu}^{\mu,h}(t)|_{\mathbb{S}^1}\right) dt = LCOT_{\mu,h}(\mu, \nu).$$

*In particular,*

$$COT_2(\mu, \nu) = \|\widehat{\nu}\|_{L^2(\mathbb{S}^1)}^2 = \|\widehat{\nu} - \widehat{\mu}\|_{L^2(\mathbb{S}^1)}^2 = LCOT(\mu, \nu).$$

**Proposition 3.5** (Invertibility of the LCOT-Embedding.). *Let $\mu \in \mathcal{P}(\mathbb{S}^1)$ be absolutely continuous with respect to the Lebesgue measure on $\mathbb{S}^1$, $h : \mathbb{R} \to \mathbb{R}_+$ a strictly convex increasing function, and let $\nu \in \mathcal{P}(\mathbb{S}^1)$. Then,*

$$\nu = (\widehat{\nu}^{\mu,h} + \mathrm{id})_{\#}\mu.$$

We refer to Proposition A.2 in the Appendix for more properties of the LCOT-Embedding.

**Theorem 3.6.** *Let $\mu \in \mathcal{P}(\mathbb{S}^1)$ be absolutely continuous with respect to the Lebesgue measure on $\mathbb{S}^1$, and let $h(x) = |x|^p$, for $1 < p < \infty$. Then $LCOT_{\mu,p}(\cdot, \cdot)^{1/p}$ is a distance on $\mathcal{P}(\mathbb{S}^1)$. In particular, $LCOT(\cdot, \cdot)^{1/2}$ is a distance on $\mathcal{P}(\mathbb{S}^1)$.*

### 3.2 LCOT INTERPOLATION BETWEEN CIRCULAR MEASURES

Given a COT Monge map and the LCOT embedding, we can compute a linear interpolation between circular measures (refer to Wang et al. (2013) for a similar approach on the Euclidean setting). First, for arbitrary measures $\sigma, \nu \in \mathcal{P}(\mathbb{S}^1)$ the COT interpolation can be written as:

$$\rho_t^{COT} := ((1-t)\mathrm{id} + tM_\sigma^\nu)_{\#}\,\sigma, \qquad t \in [0,1]. \tag{21}$$

Similarly, for a fixed reference measure $\mu \in \mathcal{P}(\mathbb{S}^1)$, we can write the LCOT interpolation as:

$$\rho_t^{LCOT} := ((1-t)(\widehat{\sigma} + \mathrm{id}) + t(\widehat{\nu} + \mathrm{id}))_{\#}\,\mu, \qquad t \in [0,1], \tag{22}$$

where we have $\rho_{t=0}^{COT} = \rho_{t=0}^{LCOT} = \sigma$ and $\rho_{t=1}^{COT} = \rho_{t=1}^{LCOT} = \nu$. In Figure 3, we show such interpolations between the reference measure $\mu$ and two arbitrary measures $\nu_1$ and $\nu_2$ for COT and LCOT. As can be seen, the COT and LCOT interpolations between $\mu$ and $\nu_i$s coincide (by definition), while the interpolation between $\nu_1$ and $\nu_2$ is different for the two methods. We also provide an illustration of the logarithmic and exponential maps to, and from, the LCOT embedding.

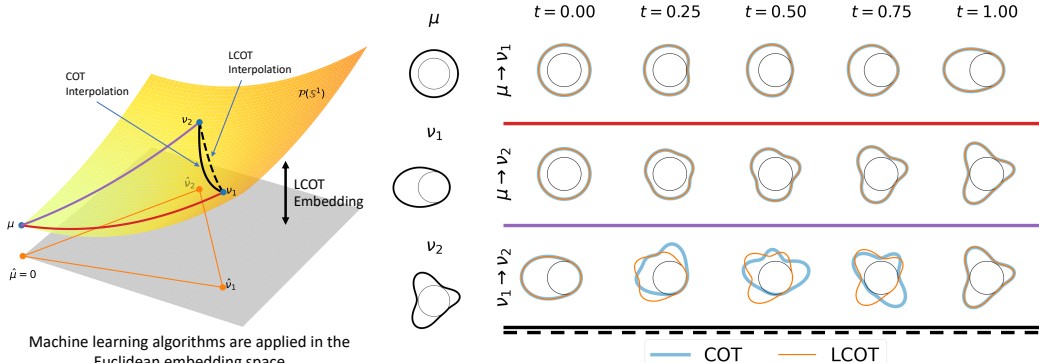

Figure 3: Left: Illustration of the LCOT embedding, the linearization process (logarithmic map), and measure interpolations. Right: Pairwise interpolations between reference measure $\mu$ and measures $\nu_1$ and $\nu_2$, using formulas in equation 21 (COT) and equation 22 (LCOT).

We refer to Appendix A.5.1 for a real data application.

### 3.3 TIME COMPLEXITY OF LINEAR COT DISTANCE BETWEEN DISCRETE MEASURES

According to (Delon et al., 2010, Theorem 6.2), for discrete measures $\nu_1, \nu_2$ with $N_1, N_2$ sorted points, the *binary search* algorithm requires $\mathcal{O}((N_1 + N_2)\log(1/\epsilon))$ computational time to find an $\epsilon$-approximate solution for $\alpha_{\nu_1, \nu_2}$. If $M$ is the least common denominator for all probability masses, an exact solution can be obtained in $\mathcal{O}((N_1 + N_2)\ln M)$. Then, for a given $\epsilon > 0$ and $K$ probability measures, $\{\nu_k\}_{k=1}^K$, each with $N$ points, the total time to pairwise compute the COT distance is $\mathcal{O}(K^2 N \ln(1/\epsilon))$. For LCOT, when the reference $\mu$ is the Lebesgue measure, the optimal $\alpha_{\mu,\nu_k}$ has a closed-form solution (see equation 10) and the time complexity for computing the LCOT embedding via equation 19 is $\mathcal{O}(N)$. The LCOT distance calculation between $\nu_i$ and $\nu_j$ according to equation 20 requires $\mathcal{O}(N)$ computations. Hence, the total time for pairwise LCOT distance computation between $K$ probability measures, $\{\nu_k\}_{k=1}^K$, each with $N$ points, would be $\mathcal{O}(K^2 N + KN)$. See Appendix A.3 for further explanation.

To verify these time complexities, we evaluate the computational time for COT and LCOT algorithms and present the results in Figure 4. We generate $K$ random discrete measures, $\{\nu_k\}_{k=1}^K$, each with $N$ samples, and for the reference measure, $\mu$, we choose: 1) the uniform discrete measure, and 2) a random discrete measure, both with $N_0 = N$ samples. To calculate $\alpha_{\mu,\nu_k}$, we considered the two scenarios, using the binary search Delon et al. (2010) for the non-uniform reference, and using equation 10 for the uniform reference. We labeled them as, "uniform ref." and "non-uniform ref." Then, in our first experiment, we set $K = 2$ and measured the wall-clock time for calculating COT and LCOT while varying $N \in \{100, 200, \ldots, 20000\}$. For our second experiment, and for $N \in \{500, 1000, 5000\}$, we vary $K \in \{2, 4, 6, \ldots, 64\}$ and measure the total time for calculating pairwise COT and LCOT distances. The computational benefit of LCOT is evident from Figure 4.

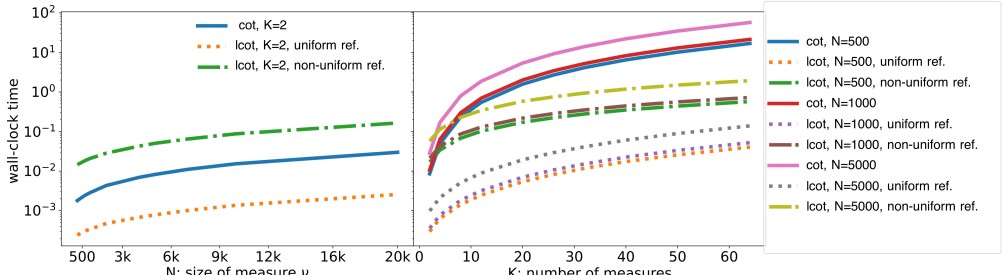

Figure 4: Computational time analysis of COT and LCOT, for pairwise comparison of $K$ discrete measures, each with $N$ samples. Left: Wall-clock time in seconds for $K = 2$ and $N \in \{100, 200, \ldots, 20000\}$. Right: Wall-clock time in seconds for $N \in \{500, 1000, 5000\}$, and $K \in \{2, 4, 6, \ldots, 64\}$. Solid lines are COT, dotted are LCOT with a uniform reference and dashdotted are LCOT with a non-uniform reference.

## 4 Experiments

To better understand the geometry induced by the LCOT metric, we perform Multidimensional Scaling (MDS) (Kruskal, 1964) on a family of densities, where the discrepancy matrices are computed using LCOT, COT, OT (with a fixed cutting point), and the Euclidean distance.

**Experiment 1.** We generate three families of circular densities, calculate pairwise distances between them, and depict their MDS embedding in Figure 5. In short, the densities are chosen as follows; we start with two initial densities: (1) a von Mises centered at the south pole of the circle ($\mu$=0.5), (2) a bimodal von Mises centered at the east ($\mu$=0.25) and west ($\mu$=0.75) ends of the circle. Then, we create 20 equally distant circular translations of each of these densities to capture the geometry of the circle. Finally, we parametrize the COT geodesic between the initial densities and generate 20 extra densities on the geodesic. Figure 5 shows these densities in green, blue, and red, respectively. The representations given by the MDS visualizations show that LCOT and COT capture the geometry of the circle coded in the translation property in an intuitive fashion. In contrast, OT and Euclidean distances do not capture the underlying geometry of the problem.

**Experiment 2.** To assess the separability properties of the LCOT embedding, we follow a similar experiment design as in Landler et al. (2021). We consider six groups of circular density functions as in the third row of Figure 5: unimodal von Mises (axial: 0°), wrapped skew-normal, symmetric bimodal von Mises (axial: 0° and 180°), asymmetric bimodal von Mises (axial: 0° and 120°), symmetric trimodal von Mises (axial: 0°, 120° and 240°), asymmetric trimodal von Mises (axial: 0°, 240° and 225°). We assign a von Mises distribution with a small spread ($\kappa = 200$) to each distribution's axis/axes to introduce random perturbations of these distributions. We generate 20 sets of simulated densities and sample each with 50-100 samples. Following the computation of pairwise distances among the sets of samples using LCOT, COT, OT, and Euclidean methods, we again employ MDS to visualize the separability of each approach across the six circular density classes mentioned above. The outcomes are presented in the bottom row of Figure 5. It can be seen that LCOT stands out for its superior clustering outcomes, featuring distinct boundaries between the actual classes, outperforming the other methods.

**Experiment 3.** In our last experiment, we consider the calculation of the barycenter of circular densities. Building upon Experiments 1 and 2, we generated unimodal, bimodal, and trimodal von Mises distributions. For each distribution's axis/axes, we assigned a von Mises distribution with a

small spread ($\kappa = 200$) to introduce random perturbations. These distributions are shown in Figure 6 (left). Subsequently, we computed both the Euclidean average of these densities and the LCOT barycenter. Notably, unlike COT, the invertible nature of the LCOT embedding allows us to directly calculate the barycenter as the inverse of the embedded distributions' average (see Appendix A.4). The resulting barycenters are illustrated in Figure 6. As observed, the LCOT method accurately recovers the correct barycenter without necessitating additional optimization steps.

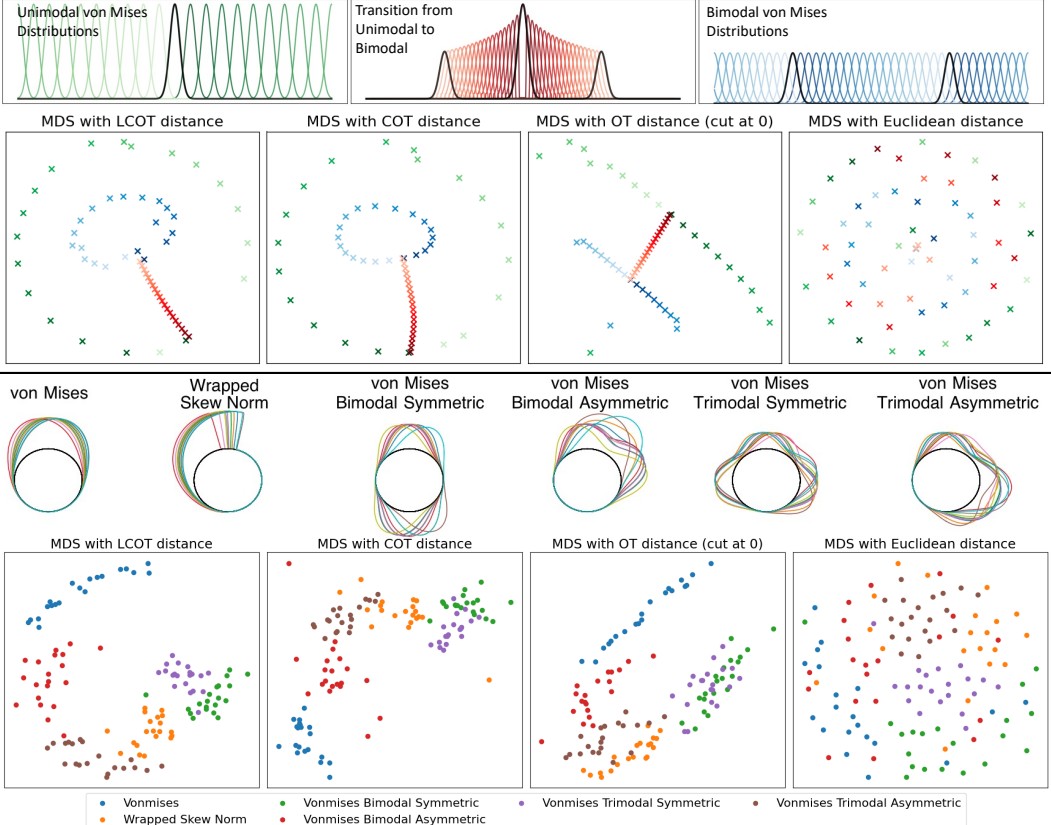

Figure 5: MDS for embedding classes of probability densities into an Euclidean space of dimension 2 where the original pair-wise distances (COT-distance, LOT-distance, Euclidean or $L^2$-distance) are preserved as well as possible.

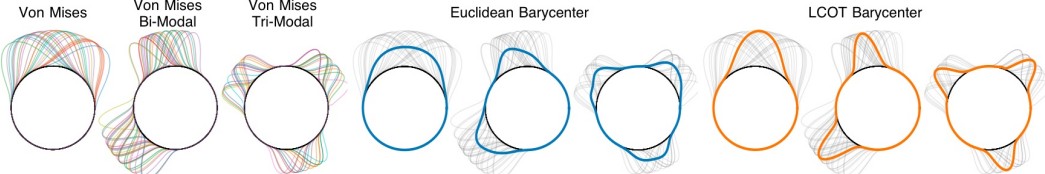

Figure 6: The LCOT barycenter compared to the Euclidean mean.

## 5 CONCLUSION AND DISCUSSION

In this paper, we present the Linear Circular Optimal Transport (LCOT) discrepancy, a new metric for circular measures derived from the Linear Optimal Transport (LOT) framework Wang et al. (2013); Kolouri et al. (2016); Park et al. (2018); Cai et al. (2022); Aldroubi et al. (2022); Moosmüller & Cloninger (2023). The LCOT offers 1) notable computational benefits over the COT metric, particularly in pairwise comparisons of numerous measures, and 2) a linear embedding where the $\| \cdot \|_{L^2(\mathbb{S}^1)}$ between embedded distributions equates to the LCOT metric. We consolidated scattered results on circular OT into Theorem 2.5 and introduced the LCOT metric and embedding, validating LCOT as a metric in Theorem 3.6. In Section 3.3, we assess LCOT's computational complexity for pairwise comparisons of $K$ circular measures, juxtaposing it with COT. We conclude by showcasing LCOT's empirical strengths via MDS embeddings on varied circular densities using different metrics.

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

# A APPENDIX

## A.1 PROOFS

*Proof of Proposition 2.3.* The proof of Proposition 2.3 is provided in Delon et al. (2010) for the optimal coupling for any pair of probability measures on $\mathbb{S}^1$. For the particular and enlightening case of discrete probability measures on $\mathbb{S}^1$, we refer the reader to Rabin et al. (2011).

For completeness, notice that the relation between $x_0$ and $\alpha$ hols by changing variables, using 1-periodicity of $\mu$ and $\nu$ and Definition 2.2 (see also Bonet et al. (2023, Proposition 1)):

$$\int_0^1 h(|F_{\mu,x_0}^{-1}(x) - F_{\nu,x_0}^{-1}(x)|_{\mathbb{R}})\, dx$$

$$= \int_0^1 h(|(F_\mu(\cdot + x_0) - F_\mu(x_0))^{-1}(x) - (F_\nu(\cdot + x_0) - F_\nu(x_0))^{-1}(x)|_{\mathbb{R}})\, dx$$

$$= \int_0^1 h(|(F_\mu - F_\mu(x_0))^{-1}(x) - (F_\nu - F_\nu(x_0))^{-1}(x)|_{\mathbb{R}})\, dx$$

$$= \int_0^1 h(|F_\mu^{-1}(x + F_\mu(x_0)) - F_\nu^{-1}(x + F_\nu(x_0))|_{\mathbb{R}})\, dx$$

$$= \int_0^1 h(|F_\mu^{-1}(x + \underbrace{F_\mu(x_0) - F_\nu(x_0)}_{\alpha}) - F_\nu^{-1}(x)|_{\mathbb{R}})\, dx$$

In particular, if $h(x) = |x|^2$, and $\mu = Unif(\mathbb{S}^1)$, then

$$COT_2(\mu, \nu) = \inf_{\alpha \in \mathbb{R}} \int_0^1 |F_\mu^{-1}(x + \alpha) - F_\nu^{-1}(x)|_{\mathbb{R}}^2\, dx$$

$$= \inf_{\alpha \in \mathbb{R}} \int_0^1 |x + \alpha - F_\nu^{-1}(x)|^2\, dx$$

$$= \inf_{\alpha \in \mathbb{R}} \left( \int_0^1 |F_\nu^{-1}(x) - x|^2\, dx - 2\alpha \int_0^1 (F_\nu^{-1}(x) - x)\, dx + \alpha^2 \right)$$

$$= \inf_{\alpha \in \mathbb{R}} \left( \int_0^1 |F_\nu^{-1}(x) - x|^2\, dx - 2\alpha \left( \int_0^1 x\, d\nu(x) - \frac{1}{2} \right) + \alpha^2 \right)$$

$$= \inf_{\alpha \in \mathbb{R}} \left( \int_0^1 |F_\nu^{-1}(x) - x|^2\, dx - 2\alpha \left( \mathbb{E}(\nu) - \frac{1}{2} \right) + \alpha^2 \right)$$

$$= \int_0^1 |F_\nu^{-1}(x) - x|^2\, dx - 2\underbrace{\left( \mathbb{E}(\nu) - \frac{1}{2} \right)^2}_{\alpha_{\mu,\nu}}.$$

Therefore, in this case, the minimizer $\alpha_{\mu,\nu}$ of equation 8 is unique and has the closed-form $\alpha_{\mu,\nu} = \mathbb{E}(\nu) - 1/2$. $\qquad\square$

*Proof of Remark 2.4.* We will show that, *in general*, the minimizer $\alpha_{\mu,\nu}$ of equation 8 is unique. Our arguments are based on the paper Delon et al. (2010). Specifically, the role played by the function $(F^\theta)^{-1}$, where $F^\theta(x) = F(x) + \theta$ in Delon et al. (2010) is substituted by $F^{-1}(x - \alpha)$ in our case (i.e., our parameter $\alpha$ correspond to $-\theta$ in the mentioned paper).

Our hypotheses are the following:

1. Let $c(x, y) := h(|x - y|)$ for $h : \mathbb{R} \to \mathbb{R}_+$ strictly convex (for example $h(x) = |x|^p$ with $p > 1$), or, more generally, let $c : \mathbb{R} \times \mathbb{R} \to \mathbb{R}$ satisfying the *Monge condition* or the *(strict) cyclical monotonicity condition:*

$$c(u_1, v_1) + c(u_2, v_2) < c(u_1, v_2) + c(u_2, v_1) \qquad \forall u_1 < u_2, v_1 < v_2. \tag{23}$$

2. Let $\mu, \nu$ be two probability measures absolutely continuous with respect to the Lebesgue measure on $\mathbb{S}^1$.

The idea of the proof will rely on showing that the cost function

$$\text{Cost}(\alpha) := \int_0^1 c(F_\mu^{-1}(x), F_\nu^{-1}(x - \alpha)) \, dx \tag{24}$$

is strictly convex and continuous (as a function on $\alpha$), and so it has a unique global minimum (that we will denote by $\alpha_{\mu,\nu}$).

Let

$$c_{\mu,\nu}(x, y) := c(F_\mu^{-1}(x), F_\nu^{-1}(y)) \tag{25}$$

Under the above conditions, it holds that $c_{\mu,\nu}(\cdot, \cdot)$ satisfies the Monge condition equation 23. Then, to prove strictly convexity of $\text{Cost}(\alpha)$ it is sufficient to show that

$$\text{Cost}\left(\frac{\alpha' + \alpha''}{2}\right) < \frac{\text{Cost}(\alpha') + \text{Cost}(\alpha'')}{2} \tag{26}$$

Assume $\alpha' \leq \alpha''$ and let $\overline{\alpha} := \frac{\alpha' + \alpha''}{2}$. On the one hand, since

$$
\begin{aligned}
\text{Cost}(\overline{\alpha}) &= \int_0^1 c_{\mu,\nu}(x, x - \overline{\alpha}) \, dx \\
&= \int_{\alpha'' - \overline{\alpha}}^{1 + \alpha'' - \overline{\alpha}} c_{\mu,\nu}(x, x - \overline{\alpha}) \, dx \\
&= \int_0^1 c_{\mu,\nu}(y + \alpha'' - \overline{\alpha}, y - \alpha') \, dy,
\end{aligned}
$$

where in the last line we used the change of variables $y = x + \alpha'' - \overline{\alpha}$ and the fact that $2\overline{\alpha} - \alpha'' = \alpha'$. Therefore,

$$2\,\text{Cost}(\overline{\alpha}) = \int_0^1 c_{\mu,\nu}(z, z - \overline{\alpha}) \, dz + \int_0^1 c_{\mu,\nu}(z + \alpha'' - \overline{\alpha}, z - \alpha') \, dz. \tag{27}$$

On the other hand, by repeating the same idea we have,

$$
\begin{aligned}
\text{Cost}(\alpha'') &= \int_0^1 c_{\mu,\nu}(x, x - \alpha'') \, dx = \int_{\alpha'' - \overline{\alpha}}^{1 + \alpha'' - \overline{\alpha}} c_{\mu,\nu}(x, x - \alpha'') \, dx \\
&= \int_0^1 c_{\mu,\nu}(y + \alpha'' - \overline{\alpha}, y - \overline{\alpha}) \, dy,
\end{aligned}
$$

and so,

$$\text{Cost}(\alpha') + \text{Cost}(\alpha'') = \int_0^1 c_{\mu,\nu}(z, z - \alpha') \, dz + \int_0^1 c_{\mu,\nu}(z + \alpha'' - \overline{\alpha}, z - \overline{\alpha}) \, dz. \tag{28}$$

Since $u_1(z) := z < z + \alpha'' - \overline{\alpha} =: u_2(x)$ and $v_1(z) := z - \overline{\alpha} < z - \alpha' =: v_2(z)$, we have that

$$c(u_1(z), v_1(z)) + c(u_2(z), v_2(z)) < c(u_1(z), v_2(z)) + c(u_2(z), v_1(z))$$

because $c_{\mu,\nu}(\cdot, \cdot)$ satisfies Monge condition equation 23. Thus, from equation 27 and equation 28 we obtain

$$
\begin{aligned}
2\,\text{Cost}(\overline{\alpha}) &= \int_0^1 c_{\mu,\nu}(u_1(z), v_1(z)) \, dz + \int_0^1 c_{\mu,\nu}(u_2(z), v_2(z)) \, dz \\
&< \int_0^1 c_{\mu,\nu}(u_1(z), v_2(z)) \, dz + \int_0^1 c_{\mu,\nu}(u_2(z), v_1(z)) \, dz = \text{Cost}(\alpha') + \text{Cost}(\alpha''),
\end{aligned}
$$

and so equation 26 holds. The continuity of $\alpha \mapsto \text{Cost}(\alpha)$ holds as the integral in equation 24 is finite for all $\alpha$.

$\square$

**Remark A.1.** *We mention that in the case that $c(x, y) = h(x - y)$ for $h(x) = |x|$ (studied for example in Cabrelli & Molter (1998); Hundrieser et al. (2022)) we have convexity but not strictly convexity. However, the authors in Hundrieser et al. (2022) prove that a closed-formula for a minimizer $\alpha_{\mu,\nu}$ of equation 8:*

$$\alpha_{\mu,\nu} = \min \left\{ \arg\min_{u \in \mathbb{R}} \int_0^1 |(F_\mu - F_\nu)(t) - u| dt \right\}$$

*called the Level Median of the function $F_\mu - F_\nu$. To show that, it is used the fact that in this case equation 8 takes the form*

$$COT_h(\mu, \nu) = \inf_{\alpha \in \mathbb{R}} \int_0^1 |F_\mu(t) - F_\nu(t) - \alpha| \, dt.$$

*Proof of Theorem 2.5.*

1. First, we will show that the map $M_\mu^\nu$ given by equation 14 satisfies $(M_\mu^\nu)_{\#}\mu = \nu$. Here $\mu$ and $\nu$ are the extended measures form $\mathbb{S}^1$ to $\mathbb{R}$ having CDFs equal to $F_\mu$ and $F_\nu$, respectively, defined by equation 3 and 4. By choosing the system of coordinates $\widetilde{x} \in [0, 1)$ that starts at $x_{cut}$ (see Figure 7) then,

$$M_\mu^\nu(\widetilde{x}) = F_{\nu, x_{cut}}^{-1} \circ F_{\mu, x_{cut}}(\widetilde{x})$$

(see equation 12). Let $\mu_{x_{cut}}$ and $\nu_{x_{cut}}$ be the (1-periodic) measures on $\mathbb{R}$ having CDFs $F_{\mu, x_{cut}}$ and $F_{\nu, x_{cut}}$, respectively, i.e., $F_{\nu, x_{cut}}(\widetilde{x}) = \mu_{x_{cut}}([0, \widetilde{x}))$ (analogously for $\nu_{x_{cut}}$). That is, we have unrolled $\mu$ and $\nu$ from $\mathbb{S}^1$ to $\mathbb{R}$, where the origin $0 \in \mathbb{R}$ corresponds to $x_{cut} \in \mathbb{S}^1$ (see Figure 1). Thus, a classic computation yields

$$(F_{\nu, x_{cut}}^{-1} \circ F_{\mu, x_{cut}})_{\#}\mu_{x_{cut}} = (F_{\nu, x_{cut}}^{-1})_{\#} \left((F_{\mu, x_{cut}})_{\#}\mu_{x_{cut}}\right) = (F_{\nu, x_{cut}}^{-1})_{\#}\mathcal{L}_{\mathbb{S}^1} = \nu_{x_{cut}}$$

where $\mathcal{L}_{\mathbb{S}^1} = Unif(\mathbb{S}^1)$ denotes the Lebesgue measure on the circle. We used that $(F_\mu)_{\#}\mu = \mathcal{L}_{\mathbb{S}^1}$ as $\mu$ does not give mass to atoms, and so, if we change the system of coordinates we also have $(F_{\mu, x_{cut}})_{\#}\mu_{x_{cut}} = \mathcal{L}_{\mathbb{S}^1}$.

   Finally, we have to switch coordinates. Let

$$z(\widetilde{x}) := \widetilde{x} + x_{cut}$$

(that is, $z(\widetilde{x}) = x$). To visualize this, see Figure 7. It holds that

$$z_{\#}\nu_{cut} = \nu \tag{29}$$

(where we recall that $\nu$ is the extended measure form $\mathbb{S}^1$ to $\mathbb{R}$ having CDF equal to $F_\mu$ as in equation 3 and 4). Let us check this fact for intervals:

$$\begin{aligned} z_{\#}\nu_{x_{cut}}([a, b]) &= \nu_{x_{cut}}(z^{-1}([a, b])) = \nu([z^{-1}(a), z^{-1}(b)]) \\ &= \nu_{x_{cut}}([a - x_{cut}, b - x_{cut}]) \\ &= F_{\nu, x_{cut}}(b - x_{cut}) - F_{\nu, x_{cut}}(a - x_{cut}) \\ &= F_\nu(b) - F_\nu(x_{cut}) - (F_\nu(a) - F_\nu(x_{cut})) \\ &= F_\nu(b) - F_\nu(a) \\ &= \nu([a, b]). \end{aligned}$$

   Besides, it holds that

$$F_{\mu, x_{cut}}(\cdot - x_{cut})_{\#}\mu = Unif(\mathbb{S}^1), \tag{30}$$

in the sense that it is the Lebesgue measure on $\mathbb{S}^1$ extended periodically (with period 1) to the real line, which we denote by $\mathcal{L}_{\mathbb{S}^1}$. Let us sketch the proof for intervals. First, notice that $F_{\mu, x_{cut}}(x - x_{cut}) = F_\mu(x) - F_\mu(x_{cut})$ and so its inverse is $y \mapsto F_\mu^{-1}(y + x_{cut})$. Therefore,

$$\begin{aligned} (F_{\mu, x_{cut}}(\cdot - x_{cut}))_{\#}\mu([a, b]) &= \mu\left([F_\mu^{-1}(a + x_{cut}), F_\mu^{-1}(b + x_{cut})]\right) \\ &= F_\mu(F_\mu^{-1}(a + x_{cut})) - F_\mu(F_\mu^{-1}(b + x_{cut})) = b - a. \end{aligned}$$

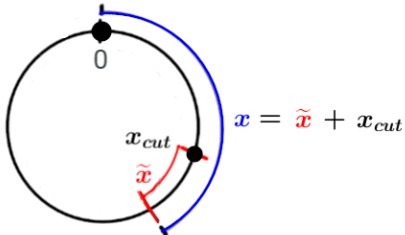

Figure 7: The unit circle (black) can be parametrized as $[0, 1)$ in many different ways. In the figure, we marked in black the North Pole as $0$. The canonical parametrization of $\mathbb{S}^1$ identifies the North Pole with $0$. Then, also in black, we pick a point $x_{cut}$. The distance in blue $x$ that starts at $0$ equals the distance in red $\tilde{x}$ that starts at $x_{cut}$ plus the corresponding starting point $x_{cut}$. This allows us to visualize the change of coordinates given by equation 13.

Finally,

$$
\begin{aligned}
(M_\mu^\nu)_{\#}\mu &= \left(F_{\nu,x_{cut}}^{-1}(F_{\mu,x_{cut}}(\cdot - x_{cut}) + x_{cut})\right)_{\#}\mu \\
&= \left(z(F_{\nu,x_{cut}}^{-1}(F_{\mu,x_{cut}}(\cdot - x_{cut}))))\right)_{\#}\mu \\
&= z_{\#}(F_{\nu,x_{cut}}^{-1})_{\#}(F_{\mu,x_{cut}}(\cdot - x_{cut}))_{\#}\mu \\
&= z_{\#}(F_{\nu,x_{cut}}^{-1})_{\#}\mathcal{L}_{\mathbb{S}^1} \qquad \text{(by equation 30)} \\
&= z_{\#}\nu_{x_{cut}} \\
&= \nu \qquad \text{(by equation 29)}.
\end{aligned}
$$

Now, let us prove that $M_\mu^\nu$ is optimal.

First, assume that $\mu$ is absolutely continuous with respect to the Lebesgue measure on $\mathbb{S}^1$ and let $f_\mu$ denote its density function. We will use the change of variables

$$
\begin{cases}
u = F_{\mu,x_{cut}}(x - x_{cut}) = F_\mu(x) - F_\mu(x_{cut}) \\
du = f_\mu(x)\, dx.
\end{cases}
$$

So,

$$
\begin{aligned}
\int_0^1 h(|M_\mu^\nu(x) - x|_{\mathbb{R}})\, d\mu(x) &= \int_0^1 h(|F_{\nu,x_{cut}}^{-1}(F_{\mu,x_{cut}}(x - x_{cut})) - (x - x_{cut})|_{\mathbb{R}})\, \underbrace{f_\mu(x)dx}_{d\mu(x)} \\
&= \int_{-x_{cut}}^{1-x_{cut}} h(|F_{\nu,x_{cut}}^{-1}(u) - F_{\mu,x_{cut}}^{-1}(u)|_{\mathbb{R}})\, du \\
&= \int_0^1 h(|F_{\nu,x_{cut}}^{-1}(u) - F_{\mu,x_{cut}}^{-1}(u)|_{\mathbb{R}})\, du \\
&= COT_h(\mu, \nu).
\end{aligned}
$$

Now, let us do the proof in general:

$$
\begin{aligned}
\int_0^1 h(|M_\mu^\nu(x) - x|_{\mathbb{R}})\, d\mu(x) &= \int_0^1 h(|F_{\nu,x_{cut}}^{-1}(F_{\mu,x_{cut}}(x - x_{cut})) - (x - x_{cut})|_{\mathbb{R}})\, d\mu(x) \\
&= \int_0^1 h(|F_{\nu,x_{cut}}^{-1}(y) - F_{\mu,x_{cut}}^{-1}(y)|_{\mathbb{R}})\, d(F_{\mu,x_{cut}}(\cdot - x_{cut}))_{\#}\mu(y) \\
&= \int_0^1 h(|F_{\nu,x_{cut}}^{-1}(u) - F_{\mu,x_{cut}}^{-1}(u)|_{\mathbb{R}})\, du \\
&= COT_h(\mu, \nu).
\end{aligned}
$$

In the last equality we have used that $F_{\mu,x_{cut}}(\cdot - x_{cut})_{\#}\mu$ is the Lebesgue measure (see equation 30).

2. Using the definition of the generalized inverse (quantile function), we have

$$
\begin{aligned}
M_\mu^\nu(t) &= F_{\nu,x_{cut}}^{-1}(F_{\mu,x_{cut}}(x - x_{cut})) + x_{cut} \\
&= \inf\{x' : F_{\nu,x_{cut}}(x') > F_{\mu,x_{cut}}(x - x_{cut})\} + x_{cut} \\
&= \inf\{x' : F_\nu(x' + x_{cut}) - F_\nu(x_{cut}) > F_\mu(x) - F_\mu(x_{cut})\} + x_{cut} \\
&= \inf\{x' : F_\nu(x' + x_{cut}) > F_\mu(x) - F_\mu(x_{cut}) + F(x_{cut})\} + x_{cut} \\
&= \inf\{x' : F_\nu(x' + x_{cut}) > F_\mu(x) - \alpha_{\mu,\nu}\} + x_{cut} \\
&= \inf\{y - x_{cut} : F_\nu(y) > F_\mu(x) - \alpha_{\mu,\nu}\} + x_{cut} \\
&= \inf\{y : F_\nu(y) > F_\mu(x) - \alpha_{\mu,\nu}\} + x_{cut} - x_{cut} \\
&= F_\nu^{-1}(F_\mu(x) - \alpha_{\mu,\nu}).
\end{aligned}
$$

3. This part follows from the previous item as the right-hand side of equation 16 does not depend on any minimizer of equation 7.

4. From (McCann, 2001, Theorem 13), there exists a unique optimal Monge map for the optimal transport problem on the unit circle. Therefore, by using Remark 2.4, $M_\mu^\nu$ is the unique optimal transport map from $\mu$ to $\nu$. For the quadratic case $h(x) = |x|^2$, we refer for example to Santambrogio (2015, Th. 1.25, Sec. 1.3.2)). Moreover, in this particular case, there exists a function $\varphi$ such that $M_\mu^\nu(x) = x - \nabla\varphi(x)$, where $\varphi$ is a *Kantorovich potential* (that is, a solution to the dual optimal transport problem on $\mathbb{S}^1$) and the sum is modulo $\mathbb{Z}$.

5. The identity $(M_\mu^\nu)^{-1} = (M_\nu^\mu)$ holds from the symmetry of the cost equation 5 that one should optimize. Also, it can be verified using equation 16 and the fact that from equation 9 $\alpha_{\mu,\nu} = -\alpha_{\nu,\mu}$:

$$
\begin{aligned}
M_\nu^\mu \circ M_\mu^\nu(x) &= F_\mu^{-1}\left(F_\nu(F_\nu^{-1}(F_\mu(x) - \alpha_{\mu,\nu})) - \alpha_{\nu,\mu}\right) \\
&= F_\mu^{-1}\left(F_\mu(x) - \alpha_{\mu,\nu} + \alpha_{\mu,\nu}\right) = x.
\end{aligned}
$$

$\square$

**Proposition A.2** (Properties of the LCOT-Embedding). *Let $\mu \in \mathcal{P}(\mathbb{S}^1)$ be absolutely continuous with respect to the Lebesgue measure on $\mathbb{S}^1$, and let $\nu \in \mathcal{P}(\mathbb{S}^1)$.*

1. *$\widehat{\mu}^{\mu,h} \equiv 0$.*

2. *$\widehat{\nu}^{\mu,h}(x) \in [-0.5, 0.5]$   for every $x \in [0, 1)$.*

3. *Let $\nu_1, \nu_2 \in \mathcal{P}(\mathbb{S}^1)$ with $\nu_1$ that does not give mass to atoms, then the map*

$$
M := (\widehat{\nu_2}^{\mu,h} - \widehat{\nu_1}^{\mu,h}) \circ ((\widehat{\nu_1}^{\mu,h} + \mathrm{id})^{-1}) + \mathrm{id}, \tag{31}
$$

*satisfies $M_\#\nu_1 = \nu_2$ (however, it is not necessarily an optimal circular transport map).*

*Proof of Proposition A.2.*

1. It trivially holds that the optimal Monge map from the distribution $\mu$ to itself is the identity id, or equivalently, that the optimal displacement is zero for all the particles.

2. It holds from the fact of being the optimal displacement, that is,

$$
COT_h(\mu, \nu) = \inf_{M : M_\#\mu=\nu} \int_{\mathbb{S}^1} h(|M(x) - x|_{\mathbb{S}^1})\, d\mu(x) = \int_{\mathbb{S}^1} h(|\widehat{\nu}^{\mu,h}(x)|_{\mathbb{S}^1})\, d\mu(x),
$$

and from the fact that $|z|_{\mathbb{S}^1}$ is at most 0.5.

3. We will use that $\widehat{\nu}^{\mu,h} = M_\mu^\nu - \mathrm{id}$, and that $(M_\mu^\nu)^{-1} = M_\nu^\mu$:

$$M(x) = (\widehat{\nu_2}^{\mu,h} - \widehat{\nu_1}^{\mu,h}) \circ M_{\nu_1}^\mu(x) + x$$
$$= (M_\mu^{\nu_2} - M_\mu^{\nu_1}) \circ M_{\nu_1}^\mu(x) + x$$
$$= M_\mu^{\nu_2} \circ M_{\nu_1}^\mu(x) - x + x$$
$$= M_\mu^{\nu_2} \circ M_{\nu_1}^\mu(x).$$

Finally, notice that

$$(M_\mu^{\nu_2} \circ M_{\nu_1}^\mu)_\# \nu_1 = (M_\mu^{\nu_2})_\# \left((M_{\nu_1}^\mu)_\# \nu_1\right) = (M_\mu^{\nu_2})_\# \mu = \nu_2.$$

$\square$

Now, we will proceed to prove Theorem 3.6. By having this result, it is worth noticing that $LCOT_{\mu,p}(\cdot, \cdot)^{1/p}$ endows $\mathcal{P}(\mathbb{S}^1)$ with a metric-space structure. The proof is based on the fact that we have introduced an explicit embedding and then we have considered an $L^p$-distance. It will follow that we have defined a kernel distance (that is in fact positive semidefinite).

*Proof of Theorem 3.6.* From equation 20, it is straightforward to prove the symmetric property and non-negativity.

If $\nu_1 = \nu_2$, by the uniqueness of the optimal COT map (see Theorem 2.5, Part 3), we have $\widehat{\nu_1}^{\mu,h} = \widehat{\nu_2}^{\mu,h}$. Thus, $LCOT_{\mu,h}(\nu_1, \nu_2) = 0$.

For the reverse direction, if $LCOT_{\mu,h}(\nu^1, \nu^2) = 0$, then

$$h(\min_{k \in \mathbb{Z}} \{|\widehat{\nu_1}^{\mu,h}(x) - \widehat{\nu_2}^{\mu,h}(x) + k|\}) = 0 \qquad \mu - \text{a.s.}$$

Thus,

$$\widehat{\nu_1}^{\mu,h}(x) \equiv_1 \widehat{\nu_2}^{\mu,h}(x) \qquad \mu - \text{a.s.}$$

(where $\equiv_1$ stands for the equality modulo $\mathbb{Z}$). That is,

$$M_\mu^{\nu_1}(x) = \widehat{\nu_1}^{\mu,h}(x) + x \equiv_1 \widehat{\nu_2}^{\mu,h}(x) + x = M_\mu^{\nu_2}(x) \qquad \mu \text{ a.s.}$$

Let $S \subseteq [0,1)$ denote the set of $x$ such that the equation above holds, we have $\mu(S) = 1, \mu(\mathbb{S}^1 \setminus S) = 0$. Equivalently, for any (measurable) $B \subseteq \mathbb{S}^1$, $\mu(B \cap S) = \mu(B)$. Pick any Borel set $A \subseteq \mathbb{S}^1$, we have:

$$\nu_1(A) = \mu\left((M_\mu^{\nu_1})^{-1}(A)\right)$$
$$= \mu\left((M_\mu^{\nu_1})^{-1}(A) \cap S\right)$$
$$= \mu\left((M_\mu^{\nu_2})^{-1}(A) \cap S\right)$$
$$= \mu((M_\mu^{\nu_2})^{-1}(A))$$
$$= \nu_2(A) \tag{32}$$

where the first and last equation follows from the fact $M_\mu^{\nu_1}, M_\mu^{\nu_2}$ are push forward mapping from $\mu$ to $\nu_1, \nu_2$ respectively.

Finally, we verify the triangular inequality. Here we will use that $h(x) = |x|^p$, for $1 \le p < \infty$. Let $\nu_1, \nu_2, \nu_3 \in \mathcal{P}(\mathbb{S}^1)$,

$$LCOT_{\mu,p}(\nu_1, \nu_2)^{1/p} = \left(\int_0^1 (|\widehat{\nu_1}(t) - \widehat{\nu_2}(t)|_{\mathbb{S}^1})^p \, d\mu(t)\right)^{1/p}$$

$$\le \left(\int_0^1 (|\widehat{\nu_1}(t) - \widehat{\nu_3}(t)|_{\mathbb{S}^1} + |\widehat{\nu_3}(t) - \widehat{\nu_2}(t)|_{\mathbb{S}^1})^p \, d\mu(t)\right)^{1/p}$$

$$\le \left(\int_0^1 |\widehat{\nu_1}(t) - \widehat{\nu_3}(t)|_{\mathbb{S}^1}^p \, d\mu(t)\right)^{1/p} + \left(\int_0^1 |\widehat{\nu_3}(t) - \widehat{\nu_2}(t)|_{\mathbb{S}^1}^p \, d\mu(t)\right)^{1/p}$$

$$= LCOT_{\mu,p}(\nu_1, \nu_3)^{1/p} + LCOT_{\mu,p}(\nu_2, \nu_3)^{1/p}$$

where the last inequality holds from Minkowski inequality. $\square$

## A.2 Understanding the relation between the minimizers of equation 7 and equation 8

We briefly revisit the discussion in Section equation 2.3.1, specifically in Remark equation 2.4, concerning the optimizers $x_{cut}$ and $\alpha_{\mu,\nu}$ of equation 7 and equation 8, respectively.

Assuming minimizers exist for equation 7 and equation 8, we first explain why we adopt the terminology *"cutting point"* ($x_{cut}$) for a minimizer of equation 7 and not for the minimizer $\alpha_{\mu,\nu}$ of equation 8. On the one hand, the cost function presented in 7 is given by

$$\text{Cost}(x_0) := \int_0^1 h(|F_{\mu,x_0}^{-1}(x) - F_{\nu,x_0}^{-1}(x)|_{\mathbb{R}}) \, dx. \tag{33}$$

We seek to minimize over $x_0 \in [0,1) \sim \mathbb{S}^1$, aiming to find an optimal $x_0$ that affects both CDFs $F_\mu$ and $F_\nu$. By looking at the cost 33, for each fixed $x_0$, we change the system of reference by adopting $x_0$ as the origin. Then, once an optimal $x_0$ is found (called $x_{cut}$), it leads to the optimal transportation displacement, providing a change of coordinates to unroll the CDFs of $\mu$ and $\nu$ into $\mathbb{R}$ and allowing the use the classical Optimal Transport Theory on the real line (see Section equation 2.3.2 and the proofs in Appendix equation A.1). On the other hand, the cost function in 8 is

$$\text{Cost}(\alpha) := \int_0^1 h(|F_\mu^{-1}(x+\alpha) - F_\nu^{-1}(x)|_{\mathbb{R}}) \, dx,$$

and the minimization runs over every real number $\alpha$. Here, the shift by $\alpha$ affects only one of the CDFs, not both. Therefore, it will not allow for a consistent change in the system of reference. This is why we do not refer to $\alpha$ as a cutting point in this paper, but we do refer to the minimizer of equation 7 as $x_{cut}$.

Finally, Figure 8 below is meant to provide a visualization of Remark 2.4, that is, to show through an example that, when minimizers for 7 and equation 8 do exist, while one could have multiple minimizers of 7, the minimizer of 8 is unique.

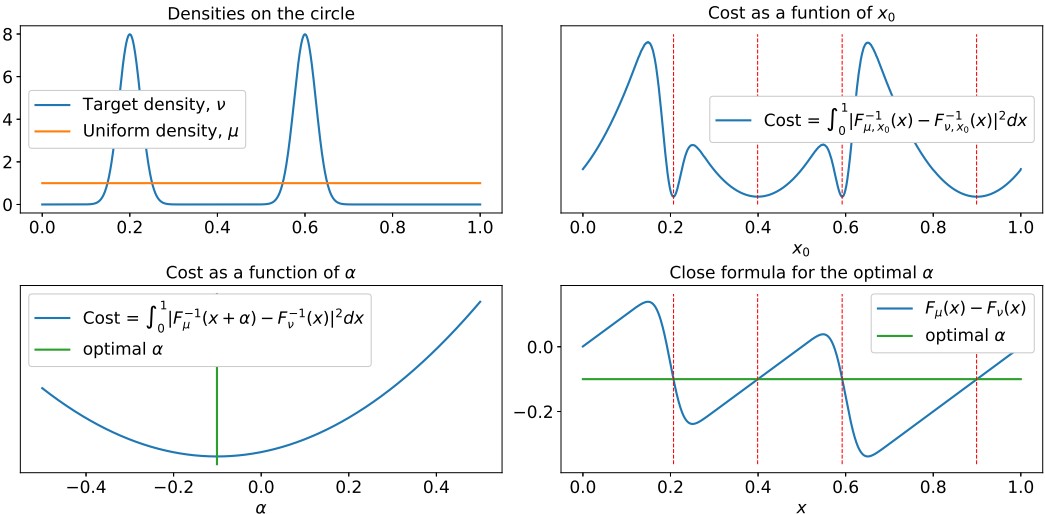

Figure 8: Top left: Uniform density, $\mu$, and a random target density $\nu$ on $\mathbb{S}^1$. Top right: The circular transportation cost $\int_0^1 |F_{\mu,x_0}^{-1}(x) - F_{\nu,x_0}^{-1}(x)|^2 \, dx$ is depicted as a function of the cut, $x_0$, showing that the optimization in equation 7 can have multiple minimizers. Bottom right: Following equation 9, we depict the difference between the two CDFs, $F_\mu(x) - F_\nu(x)$, for each $x \in [0,1) \sim \mathbb{S}^1$. As can be seen, for the optimal cuts (dotted red lines), the difference is constant, indicating that the optimal $\alpha$ for equation 8 is unique. Bottom left: The optimizer for the circular transportation cost in equation 8 is unique, and given that $\mu$ is the uniform measure, it has a closed-form solution $\mathbb{E}(\nu) - \frac{1}{2}$.

## A.3 TIME COMPLEXITY OF LINEAR COT

In this section, we assume that we are given discrete or empirical measures.[2]

First, we mention that according to (Delon et al., 2010, Section 6), given two non-decreasing step functions $F$ and $G$ represented by

$$[\,[x_1, \ldots, x_{N_1}], \; [F(x_1), \ldots, F(x_{N_1})]\,]\quad \text{and}\quad [\,[y_1, \ldots, y_{N_2}], \; [G(y_1), \ldots, G(y_{N_2})]\,],$$

the computation of an integral of the form

$$\int c(F^{-1}(x), G^{-1}(x))\,dx$$

requires $\mathcal{O}(N_1 + N_2)$ evaluations of a given cost function $c(\cdot, \cdot)$.

Now, by considering the reference measure $\mu = Unif(\mathbb{S}^1)$ we will detail our algorithm for computing $LCOT(\nu_1, \nu_2)$. Let us assume that $\nu_1, \nu_2$ are two discrete probability measures on $\mathbb{S}^1$ having $N_1$ and $N_2$ masses, respectively. We represent these measures $\nu_i = \sum_{j=1}^{N_i} m_j^i \delta_{x_j^i}$ (that is, $\nu_1$ has mass $m_j^1$ at location $x_j^1$ for $j = 1, \ldots, N_1$, and analogously for $\nu_2$) as arrays of the form

$$\nu_i = [\,[x_1^i, \ldots, x_{N_i}^i], \; [m_1^i, \ldots, m_{N_i}^i]\,], \qquad i = 1, 2.$$

Algorithm to compute LCOT:

1. For $i = 1, 2$, compute $\alpha_{\mu,\nu_i} = \mathbb{E}(\nu_i) - 1/2$.
2. For $i = 1, 2$, represent $F_{\nu_i}(\cdot) + \alpha_{\mu,\nu_i}$ as the arrays

$$[\,[x_1^i, \ldots, x_{N_i}^i], \; [c_1^i, \ldots, c_{N_i}^i]\,]$$

   where
$$c_1^i := m_1^i + \alpha_{\mu,\nu_i}, \qquad c_j^i := c_{j-1}^i + m_j^i, \quad \text{for } j = 2, \ldots, N_i.$$

3. Use that

$$F_\nu^{-1}(x - \alpha_{\mu,\nu}) = (F_\nu(\cdot) + \alpha_{\mu,\nu})^{-1}(x),$$

   and the algorithm provided in (Delon et al., 2010, Section 6) mentioned above with $F = F_{\nu_1}(\cdot) + \alpha_{\mu,\nu_1}$ and $G = F_{\nu_2}(\cdot) + \alpha_{\mu,\nu_2}$ to compute

$$LCOT(\nu_1, \nu_2) = \|\widehat{\nu_1} - \widehat{\nu_2}\|_{L^2(\mathbb{S}^1)}^2$$

$$= \int_0^1 |\,(F_{\nu_1}^{-1}(x - \alpha_{\mu,\nu_1}) - x) - (F_{\nu_2}^{-1}(x - \alpha_{\mu,\nu_2}) - x)\,|_{\mathbb{S}^1}^2\,dx$$

$$= \int_0^1 |\underbrace{(F_{\nu_1}(\cdot) + \alpha_{\mu,\nu_1})}_{F}{}^{-1}(x) - \underbrace{(F_{\nu_2}(\cdot) + \alpha_{\mu,\nu_2})}_{G}{}^{-1}(x)|_{\mathbb{S}^1}^2\,dx$$

Each step requires $\mathcal{O}(N_1 + N_2)$ operations. Therefore, the full algorithm to compute $LCOT(\nu_1, \nu_2)$ is of order $\mathcal{O}(N_1 + N_2)$.

## A.4 LCOT BARYCENTER

Although the following definition holds for any non-atomic reference measure $\mu \in \mathcal{P}(\mathbb{S}^1)$, for simplicity, we consider the reference measure as $\mu = Unif(\mathbb{S}^1)$.

Given $N$ target measures $\nu_1, \ldots, \nu_N \in \mathcal{P}(\mathbb{S}^1)$, as $LCOT_2(\cdot, \cdot)$ is a distance, their *LCOT barycenter* is defined by the measure $\nu_b$ such that

$$\nu_b = argmin_{\nu \in \mathcal{P}(\mathbb{S}^1)} \frac{1}{N} \sum_{j=1}^N LCOT_2(\nu, \nu_j) = argmin_{\nu \in \mathcal{P}(\mathbb{S}^1)} \frac{1}{N} \sum_{j=1}^N \|\widehat{\nu} - \widehat{\nu_j}\|_{L^2(\mathbb{S}^1)}^2.$$

---

[2]It is worth mentioning that for some applications, the LCOT framework can be also used for continuous densities, as in the case of the CDT Park et al. (2018).

In the embedding space, it can be shown that the minimizer of

$$argmin_{\widehat{\nu}} \frac{1}{N} \sum_{j=1}^{N} \|\widehat{\nu} - \widehat{\nu_j}\|_{L^2(\mathbb{S}^1)}^2$$

is given by the circular mean

$$\overline{\nu}(x) := \text{circle mean}(\{\widehat{\nu_1}(x), \ldots \widehat{\nu_N}(x)\}) := \frac{1}{2\pi} \arg\tan\left(\frac{\sum_{i=1}^{N} \sin(2\pi\widehat{\nu_i}(x))}{\sum_{i=1}^{N} \cos(2\pi\widehat{\nu_i}(x))}\right).$$

For each $x \in [0, 1)$, the last value is the average of the angles $\{2\pi\widehat{\nu_1}(x), \ldots, 2\pi\widehat{\nu_N}(x)\}$, which is then normalized to fall within the range $[-0.5, 0.5]$. By using the closed formula for the inverse of the LCOT Embedding provided by Proposition 3.5, we can go back to the measure space obtaining the *LCOT barycenter* between $\nu_1, \ldots, \nu_N$ as

$$\nu_b = (\overline{\nu} + \text{id})_{\#}\mu. \tag{34}$$

In our experiments, we use the expression equation 34.

## A.5 Extra Figures and Experiments

The following Figure 9 is from an experiment analogous to Experiment 1 but for a different family of measures (Figure 9 Left). We include it to have an intuition of how the LCOT behaves under translations and dilations of an initial von Mises density.

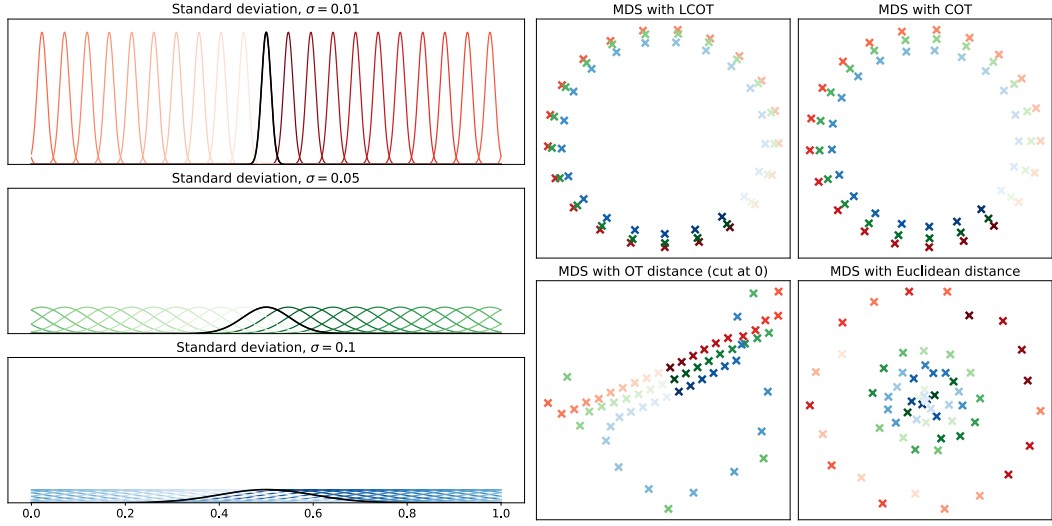

Figure 9: MDS for embedding classes of probability densities into an Euclidean space of dimension 2 where the original pair-wise distances (COT-distance, LOT-distance, Euclidean or $L^2$-distance) are preserved as well as possible.

### A.5.1 LCOT–Interpolation: Real data application

**Hue transfer experiment:** In Figures 10, 11, and 12 we interpolate the hue channel between pairs of images using LCOT interpolation (given by equation 22), and COT interpolation (given by equation 21). Given a pair of images, one is considered as the *source* and the other as the *target*. Each image of $M \times N$ pixels is represented using Hue, Saturation, and Value channels (HSV). We compute a density-normalized histogram of the Hue values across all pixels and consider this histogram as a circular density. Each bin represents at the same time a color variety (Hue value) and a point (angle) in the circle. Thus, displacements in the circle correspond to color conversions. Each interpolation (LCOT / COT) provides a curve of color conversions parametrized between $t = 0$ (source) and $t = 1$ (target). For three different pairs of images, Figures 10, 11, and 12 depict color-converted images using steps $t = 0.25, 0.5, 0.75$ for each interpolation type.

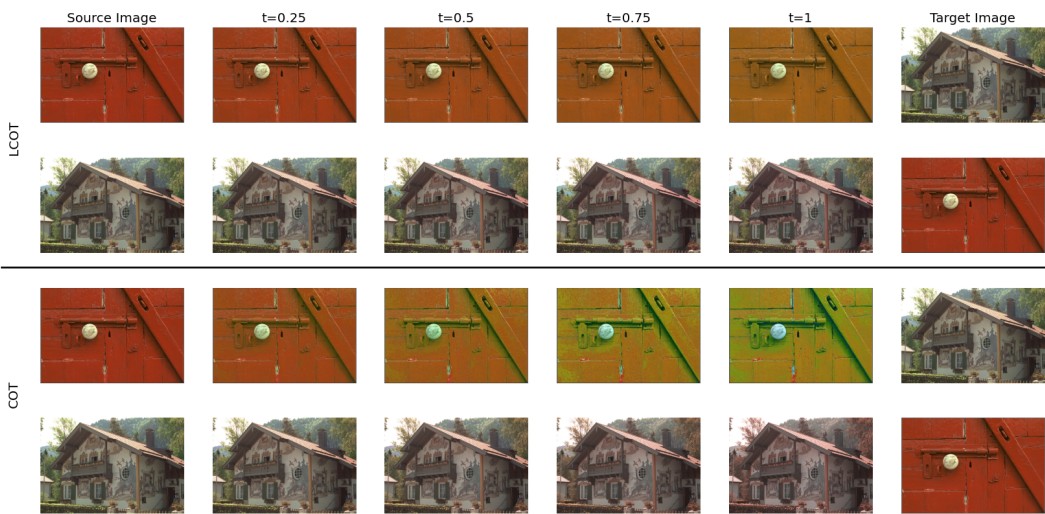

Figure 10: LCOT and COT color interpolations.

Figure 11: LCOT and COT color interpolations.

Figure 12: LCOT and COT color interpolations.

### A.5.2 LCOT FOR HUE-BASED IMAGE RETRIEVAL

Given a data set of $N = 100$ flower images represented using hue, saturation, and value channels (HSV), we use the hue channel for image retrieval. For this, the hue information of an image is used as a primary feature for searching. We extract the hue component from the images of the data set and compute density-normalized histograms of the hue values across all pixels. We consider these histograms as circular densities $\{\nu_i\}_{i=1}^N$ (similarly to A.5.1).

For LCOT comparison and retrieval, we compute the LCOT transforms $\{\widehat{\nu_i}\}_{i=1}^N$. Given a new query image, we compute its hue histogram denoted $\sigma$. Then, we perform LCOT-matching, that is, we embed the input image in LCOT space by computing $\widehat{\sigma}$ so that we can calculate $N$ squared Euclidean distances $\|\widehat{\sigma} - \widehat{\nu_i}\|_2^2$ (i.e., we are computing $LCOT_2(\sigma, \nu_i)$ for $i = 1, \ldots, N$). We sort the obtained LCOT-distance values in ascending order. In Figure 13 we show the four closest and furthest images recovered using this technique.

In Figure 14, we repeat the same experiment but using classic COT-matching for the same data set. As before, given a query image, we first compute its hue histogram denoted by $\sigma$. Then, we perform $N$ COT-distances $COT_2(\sigma, \nu_i)$, for $i = 1, \ldots, N$, and We sort the obtained COT-distance values.

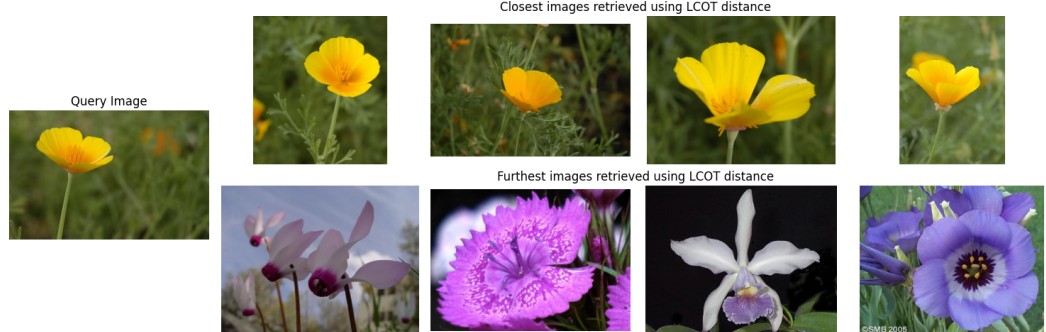

Figure 13: LCOT-approach for Hue-based image retrieval. The leftmost image is the original query image. In the upper row, we retrieve the 4 closest images in Hue space according to LCOT, while the bottom row shows the 4 furthest images with respect to LCOT-distance.

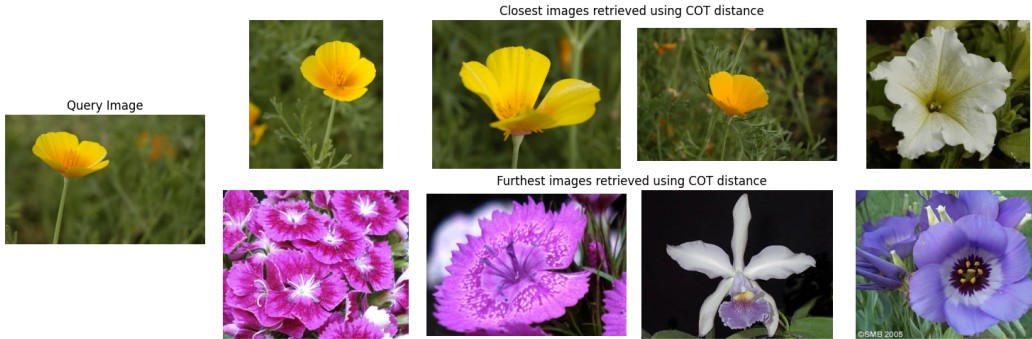

Figure 14: COT-distance for Hue-based image retrieval. The leftmost image is the original query image. In the upper row, we retrieve the 4 closest images in Hue space according to COT-distance, while the bottom row shows the 4 furthest images with respect to COT-distance.

In both figures, the retrieval of images with similar color content is evident. The advantage of using LCOT over COT is that when using COT each new image requires to solve $N$ new circular optimal transport problems whereas LCOT only requires to solve one followed by $N$ Euclidean distance calculations for comparison and sorting. For $M$ queries we have to compute $MN$ COT distances when using the COT approach ($N$ for each query) but only solve $N + M$ COT problems when using LCOT ($N$ for pre-processing the data set + one per query).

A.6 UNDERSTANDING THE EMBEDDING IN DIFFERENTIAL GEOMETRY

Our embedding $\nu \mapsto \widehat{\nu}$ as given by equation equation 17 aligns with the definition of the Logarithm function presented in (Sarrazin & Schmitzer, 2023, Definition 2.7). To be specific, for $\mu, \nu \in \mathbb{S}^1$ and the Monge mapping $M_\mu^\nu$, the Logarithm function as introduced in Sarrazin & Schmitzer (2023) is expressed as:

$$\mathcal{P}_2(\mathbb{S}^1) \ni \nu \mapsto \log_\mu^{COT}(\nu) \in L^2(\mathbb{S}^1, T\mathbb{S}^1; \mu).$$

Here, the tangent bundle of $\mathbb{S}^1$ is represented as

$$T\mathbb{S}^1 := \{(x, T_x(\mathbb{S}^1)) \,|\, x \in \mathbb{S}^1\},$$

where $T_x(\mathbb{S}^1)$ denotes the tangent space at the point $x \in \mathbb{S}^1$. The space $L^2(\mathbb{S}^1, T\mathbb{S}^1; \mu)$ is the set of vector fields on $\mathbb{S}^1$ with squared norms (based on the metric on $T\mathbb{S}^1$), that are $\mu$-integrable. The function (vector field) $\log_\mu^{COT}(\nu)$ is defined as:

$$\log_\mu^{COT}(\nu) := (\mathbb{S}^1 \ni x \mapsto (x, v_x)) \in T\mathbb{S}^1,$$

where $v_x \mapsto T_x(\mathbb{S}^1)$ is the initial velocity of the unique constant speed geodesic curve $x \mapsto T_\mu^\nu(x)$.

The relation between $\log_\mu^{COT}(\nu)$ and $\widehat{\nu}$ in equation 19 can be established as follows: For any $x$ in $\mathbb{S}^1$, the spaces $T_x(\mathbb{S}^1)$ and $\mathbb{S}^1$ can be parameterized by $\mathbb{R}$ and $[0, 1)$, respectively. Then, the unique constant speed curve $x \mapsto M_\mu^\nu(x)$ is given by:

$$x(t) := x + t(M_\mu^\nu(x) - x), \qquad \forall t \in [0, 1].$$

Then, the initial velocity is $M_\mu^\nu(x) - x$. Drawing from equation 15, Theorem 2.5, and Proposition A.2, we find $\widehat{\nu}(x) = M_\mu^\nu(x) - x$ for all $x$ in $\mathbb{S}^1$, making $\widehat{\nu}$ and $\log_\mu^{COT}(\nu)$ equivalent.

However, it is important to note that while $\log_\mu^{COT}$ is defined for a generic (connected, compact, and complete[3]) manifold, it does not provide a concrete computational method for the embedding $\log_\mu^{COT}$. Our focus in this paper is on computational efficiency, delivering a closed-form formula.

Regarding the embedding space, in Sarrazin & Schmitzer (2023), the space $L^2(\mathbb{S}^1, T\mathbb{S}^1; \mu)$ is equipped with the $L^2$, induced by $T\mathbb{S}^1$. Explicitly, for any $f$ belonging to $L^2(\mathbb{S}^1, T\mathbb{S}^1; \mu)$,

$$\|f\|^2 = \int_0^1 \|f(x)\|_x^2 dx = \int_0^1 |f(x)|^2 \, dx,$$

where $\|f(x)\|_x^2$ represents the norm square in the tangent space $T_x(\mathbb{S}^1)$ of the vector $f(x)$. By parameterizing $\mathbb{S}^1$ and $T_x(\mathbb{S}^1)$ as $[0, 1)$ and $\mathbb{R}$, respectively, this squared norm becomes $|f(x)|^2$. Consequently, $L^2(\mathbb{S}^1, T\mathbb{S}^1; \mu)$ becomes an inner product space, whereby the expression (polarization identity) $\|f + g\|^2 - \|f\|^2 - \|g\|^2$ establishes an inner product between $f$ and $g$.

However, in this paper, the introduced embedding space $L^2(\mathbb{S}^1, d\mu)$ presented in equation 18. This space uses the $L^2$-norm on the circle, defined for each $f$ in $L^2(\mathbb{S}^1, d\mu)$ as:

$$\|f\|^2_{L^2(\mathbb{S}^1; d\mu)} = \int_{\mathbb{S}^1} |f(x)|^2_{\mathbb{S}^1} \, d\mu.$$

Unlike the previous space, this does not induce an inner product (in fact, $|\cdot|_{\mathbb{S}^1}$ is not a norm). As such, throughout this paper, we term our embedding as a "linear embedding" rather than a "Euclidean embedding".

---

[3]In Sarrazin & Schmitzer (2023), the Riemannian manifold is not necessarily compact. However, the measures $\mu, \nu$ must have compact support sets. For brevity, we have slightly overlooked this difference.

