# OpenReview forum: "LCOT: Linear Circular Optimal Transport"
_ICLR.cc/2024/Conference — ICLR 2024 poster_

### Official Review · Reviewer_8oTS · 2023-10-26

**Soundness:** 4 excellent
**Presentation:** 4 excellent
**Contribution:** 3 good
**Rating:** 6
**Confidence:** 4

**Summary:**

This paper proposes a new computationally efficient metric in order to compare circular probability measures by extending the Linear Optimal Transport framework to the circle. The authors leverage a closed-form for the Monge map between the uniform distribution and another arbitrary probablity measure to provide a closed-form for the Linear Circular Optimal Transport metric with reference the uniform measure. Then, they study theoretically the metric and apply it to different experiments with a comparison with the Circular Optimal Transport distance.

**Strengths:**

Overall, the paper is nicely written and very clear. It proposes a new metric and show its benefit compared to the previous Circular Optimal Transport distances.

- Very well written and clear
- New efficient metric on the circle with a theoretical analysis
- Experiments which show the behavior of LCOT in comparison with COT: runtimes, comparison of the distances through a MDS and barycenter computation which cannot be directly done with COT.

**Weaknesses:**

- The paper is pretty incremental as it feels like a specification of the Linear OT framework in the particular case of the circle, which relies heavily on previous works such as [1, 2] for the Linear OT framework and [3] for the closed-form on the circle.
- The experiments are only on toy data.

[1] Sarrazin, Clément, and Bernhard Schmitzer. "Linearized optimal transport on manifolds." arXiv preprint arXiv:2303.13901 (2023).

[2] Wang, Wei, Dejan Slepčev, Saurav Basu, John A. Ozolek, and Gustavo K. Rohde. "A linear optimal transportation framework for quantifying and visualizing variations in sets of images." International journal of computer vision 101 (2013): 254-269.

[3] Bonet, Clément, and Paul Berg, and Nicolas Courty, and François Septier, and Lucas Drumetz, and Minh-Tan Pham. "Spherical Sliced-Wasserstein". International Conference on Learning Representations (2023).

**Questions:**

In my opinion, the paper would really benefit from an experiment on real data. For now, the experiments are informatives, but do not make me specially see in which scenario I would benefit to apply the linear framework. Thus, finding an application where the computations of multiple pairwise OT distances on the circles is required, seems to be very important in my opinion.

Some references seems to be missing. Notably, in [1, 2], derivations of the Monge map on the circle where made.

[1] Beraha, Mario, and Matteo Pegoraro. "Wasserstein Principal Component Analysis for Circular Measures." arXiv preprint arXiv:2304.02402 (2023).

[2] Quellmalz, Michael, Robert Beinert, and Gabriele Steidl. "Sliced optimal transport on the sphere." arXiv preprint arXiv:2304.09092 (2023).

---

> ### Author Response · Authors · 2023-11-22
>
> We thank the Reviewer for their time, consideration, and invaluable feedback.
>
> ## Weaknesses:
> * *The paper is pretty incremental as it feels like a specification of the Linear OT framework in the particular case of the circle, which relies heavily on previous works such as [1, 2] for the Linear OT framework and [3] for the closed-form on the circle.*
>
> **Comment**:
> Regarding the choice of focusing on the circle within the framework of Linear Optimal Transport (OT), we aimed to highlight the advantages of delving into a specific manifold. While previous works such as [1] operate in abstract manifolds (M), our decision to fixate on the circle enabled the derivation of new, concrete closed-form formulas, as evidenced in our paper. These expressions, however, hold significance within the defined manifold and may not extend beyond its boundaries.
>
> The utilization of the closed-form expression of alpha from [3], alongside insights from the works of Rabin et al. and Delon et al., indeed forms a foundation for certain aspects in section 2 of our paper. However, it is important to highlight that while some components build upon prior works, the framework as a whole is novel. We believe that the proposed comptuationally efficient framework could be of interest to the machine learning research community.
>
> We also acknowledge the influence and inspiration drawn from [2] in the development of some of the concepts and tools introduced in our work. However, it's crucial to note that while [2] predominantly deals with Euclidean domains, our focus on non-Euclidean domains presents its own set of challenges and complexities.
>
> ---------------------------------------------------
> [1] Sarrazin, Clément, and Bernhard Schmitzer. "Linearized optimal transport on manifolds." arXiv preprint arXiv:2303.13901 (2023).
>
> [2] Wang, Wei, Dejan Slepčev, Saurav Basu, John A. Ozolek, and Gustavo K. Rohde. "A linear optimal transportation framework for quantifying and visualizing variations in sets of images." International journal of computer vision 101 (2013): 254-269.
>
> [3] Bonet, Clément, and Paul Berg, and Nicolas Courty, and François Septier, and Lucas Drumetz, and Minh-Tan Pham. "Spherical Sliced-Wasserstein". International Conference on Learning Representations (2023).
>
> -------------------------------------------------------
>
> * *The experiments are only on toy data.*
>
> **Comment:** See Questions below.
>
> -------------------------------------------------------
> ## Questions:
>
> * In Appendix A.5.2 we added an application on image retrieval that requires multiple OT comparisons for each query.
>
> In a similar fashion, we also added an experiment on color interpolation between images in Appendix A.5.1 to show some non simulated data application. (However, this one does not require multiple pairwise OT distances.)
> _______________________________________________
>
> * We thank the reviewer for mentioning the works [1,2]. Unfortunately, we were unaware of these recent papers at the time of writing our paper. We have included these references in our manuscript.
>
> [1] Beraha, Mario, and Matteo Pegoraro. "Wasserstein Principal Component Analysis for Circular Measures." arXiv preprint arXiv:2304.02402 (2023).
>
> [2] Quellmalz, Michael, Robert Beinert, and Gabriele Steidl. "Sliced optimal transport on the sphere." arXiv preprint arXiv:2304.09092 (2023).

---

### Official Review · Reviewer_Hihy · 2023-10-31

**Soundness:** 3 good
**Presentation:** 3 good
**Contribution:** 2 fair
**Rating:** 6
**Confidence:** 4

**Summary:**

Standard OT metrics are well studied on data sample points having Euclidean structure, e.g. Wasserstein and Gromov-Wasserstein distances. Recently, the OT community addresses gained ample interest in OT metrics on non-Eculean spaces. This paper investigates OT between measures supported on the unit circle, aka circular probability measures. The paper starts with a background on circular OT by defining probability measures with 1-periodicity. This gives the CDF function with respect to a reference point. Proposition 2.3 is quite interesting since it expresses the COT between circular probability as an optimization problem on the real line by searching cutting points $x_{cut} \in [0,1)$. Using classical optimal transport on the real line, we can construct the Monge map between data points on the circular probability measures.

This work proposes LCOT a new discrepancy between circular probability measures, allowing to embedding the origin measures by computing their Monge map with respect to a fixed reference measure (uniform or Lebesgue measure). If the cost function is the $p$-power of the absolute value, then LCOT is a proper distance and coincides with the $L_p$ norm on the unit circle endowed with the reference measure.

**Strengths:**

- Paper easy to follow and well written.
- Proposing a novel linearization of the circular OT metric (LCOT) by embedding the origin measures via a Monge displacement with respect to reference measures.
- LCOT outperforms classical COT in terms of computational complexity.
- Extensive numerical experiments comparing COT and LCOT (with different reference measures) on simulated datasets.
- The proofs sound good to me.

**Weaknesses:**

- There is a backbone result in the paper which is the uniqueness of $\alpha_{\mu, \nu}$ (see Remark 2.4). There is an empirical showing of this uniqueness in Figure 8, Appendix A.2. However, I am still not convinced, since in Equation (8) one can get different cutting points, and using Equation (9), this may give different $\alpha_{\mu, \nu}$. Probably, the result is trivial but I think it deserves to be proved.
- As it can be seen in Figure 4, the LCOT with non-uniform reference measure is slower than the LCOT-Uniform. Is there any constant depending on the choice of the reference measure that can be added to the complexity of LCOT?
- The experiments are conducted on simulated data, I think the application of LCOT in real data in one of the detailed domains (see Paragraph 2 in the introduction) would add more importance to this approach. I am not aware of the existence of such benchmark data.

**Questions:**

### Related work
Missing related work: "The Statistics of Circular Optimal Transport" by Hundrieser et al. 2022.
### Minor Typos
- Page 6: Definition 3.2 ``(LCOT distance):``  I think should be (LCOT discrepancy)
- Page 7: Caption of Figure 4: "Left: Wall-clock time for $K=2$ and $N \in \{100, 200, \ldots, 20000\}$"
- Page 9: first line of conclusion "distance" --> "discrepancy
- Page 10: References: wasserstein --> Wasserstein

---

> ### Author Response · Authors · 2023-11-22
>
> We thank the Reviewer for their time, consideration, and invaluable feedback.
>
> ## Weaknesses:
>
> 1. We included the proof of the uniqueness of $\alpha_{\mu,\nu}$ in the Appendix. The argument is based on the paper [Delon et. al. "Fast transport optimization for Monge costs on the circle" SIAM Journal on Applied Mathematics (2010)]. The result holds when considering absolutely continuous measures $\mu$ and $\nu$, and cost $h(x-y)$ with $h$ an strictly convex function. Essentially, the idea is to prove that the integral in Equation (8), view as a function of $\alpha$, is continuous and strictly convex. Therefore, a unique global minimum is attained. Figure 8 in Appendix A.2 was meant to show this.
>
> 2. To the best of our knowledge, there is no such constant related to the reference measure that can affect the complexity of LCOT, except for the sample size. The reason the uniform case is faster is that, in this case, the optimal $\alpha_{\mu,\nu}$ has a closed form, allowing us to apply it directly to the computation of the embedding. For the non-uniform case, $\alpha_{\mu,\nu}$ is unknown, and we must use a COT solver (e.g., the binary search algorithm in [Delon et. al. "Fast transport optimization for Monge costs on the circle" SIAM Journal on Applied Mathematics (2010)]). The computational time for this is included in Figure 4. In other words, if we do not use the closed form of $\alpha_{\mu,\nu}$ for the uniform case, the computation cost would be the same as for the non-uniform case. For other reference measures $\mu$ there might be closed-form solutions for the minimizer $\alpha_{\mu,\nu}$  that we are not aware of.
>
> 3. We are not aware of the existence of such benchmarks either. However, we added an experiment on color interpolation between images in the Appendix (A.5.1) to show some non simulated data application. Also, in Appendix A.5.2, we added an application on image retrieval that requires multiple OT comparisons for each query.
>
> ## Questions:
>
> ### Related work
>
> Missing related work:
> The mentioned paper (["The Statistics of Circular Optimal Transport" by Hundrieser et al. 2022.]) was already cited in the manuscript and studied during our work. Please, check previous to last reference on page 10, and first line page 2.
>
>
> ### Minor Typos:
>  We thank the reviewer for the thorough read. We corrected all the typos.
> As for Definition 3.2, it is in general a discrepancy (as the reviewer pointed out). Nevertheless, when $h(x)=|x|^p$ for $1<p<\infty$ we proved that it is indeed a distance.

---

### Official Review · Reviewer_5tYi · 2023-11-07

**Soundness:** 2 fair
**Presentation:** 2 fair
**Contribution:** 2 fair
**Rating:** 6
**Confidence:** 3

**Summary:**

This work studied the optimal transport on the unit circle. Since in the optimal transport on the circle, $x_\text{cut}$ in Eqs. (7) and (9) is not obtained in the closed-form, the optimal transport on the circle does not have a closed-form solution. This work proposed its approximated variant, LCOT, such that LCOT has the closed-formued solution. Then, this work showed that in the special case, LCOT became equivalent to the original optimal transport on the circle (Remark 3.4) and evaluated the effectiveness of LCOT in the synthetic data.

**Strengths:**

1. In Fig. 4, this work demonstrated that the LCOT can be computed much faster than the original optimal transport on the circle, which is consistent with the theoretical analysis.
2. The relationship between the original optimal transport on the circle and the LCOT is discussed in Remark 3.4.

**Weaknesses:**

1. The intuitive interpretation of LCOT, including Eq. (17) and equations in Def. 3.2, is not clear to the reviewer.
2. The authors claimed that "invertible nature of the LCOT embedding allows us to directly calculate the barycenter" in Sec. 4, but the algorithm is missing. It is important for future research to show detailed algorithms in this paper.
3. In Fig. 4, the unit (e.g., seconds, mili seconds) is missing.

**Questions:**

1. Could the authors explain the definition of LCOT, including Eq. (17) and equations in Def. 3.2, more intuitively?
2. Remark 3.4. claimed that the COT and LCOT are equivalent. Can it be proved that the barycenter under COT and LCOT are also equivalent?

---

> ### Author Response · Authors · 2023-11-22
>
> We thank the Reviewer for their time, consideration, and invaluable feedback.
>
> ## Weaknesses:
> 1. See Question 1 below.
>
> 2. We explicitate the mentioned phrase in the Appendix A.4 "LCOT Barycenter". We thank the reviewer for this comment since now our definition is more concise.
>
> Precisely, the *LCOT barycenter* is defined as follows:
>
> Given $N$ target measures $\nu_1,\dots,\nu_N\in\mathcal{P}(\mathbb{S}^1)$, as $LCOT_2(\cdot,\cdot)$ is a distance, their *LCOT barycenter* is defined by the measure $\nu_b$ such that
> $$
>     \nu_b=argmin_{\nu\in\mathcal{P}(\mathbb{S}^1)}  \frac{1}{N}\sum_{j=1}^N LCOT_2(\nu,\nu_j)= argmin_{\nu\in\mathcal{P}(\mathbb{S}^1)}  \frac{1}{N}\sum_{j=1}^N ||\widehat{\nu}-\widehat{\nu_j}||^2_{L^2(\mathbb{S}^1)}.
> $$
> In the embedding space, it can be shown that the minimizer of
> $
>     argmin_{\widehat{\nu}}  \frac{1}{N}\sum_{j=1}^N ||\widehat{\nu}-\widehat{\nu_j}||^2_{L^2(\mathbb{S}^1)}
> $
> is given by the circular mean
> $$\overline{\nu}(x):=\text{circle mean}(\{\widehat{\nu_1}(x),\ldots \widehat{\nu_N}(x)\}):=
> \frac{1}{2\pi}\arg\tan\left({\frac{\sum_{i=1}^N\sin(2\pi\widehat{\nu_i}(x))}{\sum_{i=1}^N\cos(2\pi\widehat{\nu_i}(x))}}\right).
> $$
> For each $x\in[0,1)$, the last value is the average of the angles ${2\pi \widehat{\nu_1}(x), \ldots, 2\pi\widehat{\nu_N}(x)}$, which is then normalized to fall within the range $[-0.5, 0.5]$.
> By using the closed formula for the inverse of the LCOT Embedding provided by Proposition 3.5, we can go back to the measure space obtaining the *LCOT barycenter* between $\nu_1,\dots,\nu_N$ as
> $\nu_b=(\overline{\nu}+\mathrm{id})$#$\mu$.
>
> In our experiments, we use the last expression.
>
> 3. We included the units (*seconds*) in the caption of Figure 4.
>
>
> ## Questions:
>
> 1. Consider two circular measures $\nu_1$ and $\nu_2$ and the uniform circular measure as the reference $\mu$. Calculating COT from $\nu_i$ to $\mu$ can be intuitively thought as finding an optimal cut on the circle for $\nu_i$, cutting the circle into the line $[0,1)$ and then solving a one-dimensional OT problem in this flat space, which has a closed-form solution. This process allows us to find a Monge map (and the corresponding optimal displacement) between $\nu_i$ and $\mu$. Now, given that this mapping is invertible (i.e., for each push forward we also have a pull back), we can first push $\nu_1$ to the reference, $\mu$, and then pull back $\mu$ to $\nu_2$. This process will result in a transport map between $\nu_1$ and $\nu_2$, which is unique, but is not necessarily the optimal transport map between $\nu_1$ and $\nu_2$. Below, we describe this process more formally.
>
> The LCOT-Embedding encodes the optimal circular displacement given in Equation (15). That is, a reference measure $\mu$ is fixed and for each target measure $\nu$, the LCOT-Embedding $\widehat{\nu}^{\mu,h}(x)$ is defined as $M_\mu^\nu(x)-x$, where $M_\mu^\nu(x)$ is the optimal location to place $x$.
>
> For simplicity in the notation, let us denote $\widehat{\nu}^{\mu,h}$ by $\widehat{\nu}$.
>
> Since we have an optimal displacement for each $x$, we have that $\widehat{\nu}$ is a function. To measure the disimilarity between to target measures $\nu_1$ and $\nu_2$ in the embedding space, we use a definition that allows us to use the standard $p$-norms when $h(x)=|x|^p$. The particularity here is that these distances are for functions supported on the circle which makes the formulas more convoluted.
> The take away from Def 3.2 is that the LCOT distance between $\nu_1$ and $\nu_2$ is essentially a $p$-norm ($||\widehat{\nu}_1-\widehat{\nu}_2||_p$).
>
> 2. In general, COT and LCOT are not equivalent distances. In Remark 3.4 we pointed out that the COT cost between $\mu$ and $\nu$ can be recovered by using the LCOT distance when $\mu$ is the reference. For other pairs of measures, LCOT is still a 'transport based' distance, just not the same as COT. As a consequence, COT and LCOT barycenter are not equivalent for arbitrary sets of measures.

---

### Meta-Review · Area_Chair_Gwcb · 2023-12-02

**Metareview:**

This paper explores optimal transport (OT) metrics for circular probability measures, introducing Linear Circular Optimal Transport (LCOT) as a novel discrepancy measure. LCOT efficiently compares circular probability measures by extending the Linear Optimal Transport framework to the unit circle. The authors derive a closed-form solution for the Monge map between the uniform distribution and arbitrary probability measures, enabling a computationally efficient computation of LCOT. The metric, based on a power cost function, acts as a proper distance, coinciding with the norm on the unit circle under the reference measure. Theoretical analysis and experimental comparisons with Circular Optimal Transport distance support the efficacy of LCOT.

Overall, the closed-form variant of COT is a nice contribution for OT community. In the paper, it only compares the Linear version of COT and COT. It would be interesting to see the difference between the LCOT and Linear OT.

**Justification For Why Not Higher Score:**

All reviewers put weak acceptance to this paper and this paper is a borderline paper. Moreover, the approach is interesting, but it is a minor topic. So, a poster presentation is good for this work.

**Justification For Why Not Lower Score:**

N/A

---

### Decision · Program_Chairs · 2024-01-16

Accept (poster)